# Molecular profiling of driver events in metastatic uveal melanoma

Joakim Karlsson [1], Lisa M. Nilsson[1], Suman Mitra[1], Samuel Alsén[1], Ganesh Vilas Shelke [1], Vasu R. Sah[1], Elin M. V. Forsberg [1], Ulrika Stierner[1], Charlotta All-Eriksson[2], Berglind Einarsdottir[1], Henrik Jespersen [1], Lars Ny[1], Per Lindnér[1], Erik Larsson[3], Roger Olofsson Bagge [1] & Jonas A. Nilsson [1✉]

Metastatic uveal melanoma is less well understood than its primary counterpart, has a distinct biology compared to skin melanoma, and lacks effective treatments. Here we genomically profile metastatic tumors and infiltrating lymphocytes. *BAP1* alterations are over-represented and found in 29/32 of cases. Reintroducing a functional *BAP1* allele into a deficient patient-derived cell line, reveals a broad shift towards a transcriptomic subtype previously associated with better prognosis of the primary disease. One outlier tumor has a high mutational burden associated with UV-damage. *CDKN2A* deletions also occur, which are rarely present in primaries. A focused knockdown screen is used to investigate over-expressed genes associated withcopy number gains. Tumor-infiltrating lymphocytes are in several cases found tumor-reactive, but expression of the immune checkpoint receptors *TIM-3*, *TIGIT* and *LAG3* is also abundant. This study represents the largest whole-genome analysis of uveal melanoma to date, and presents an updated view of the metastatic disease.

[1] Sahlgrenska Cancer Center, Departments of Surgery, Oncology or Transplantation Surgery, Institute of Clinical Sciences at University of Gothenburg and Sahlgrenska University Hospital, Box 425, 40530 Gothenburg, Sweden. [2] St. Erik Eye Hospital, Polhemsgatan 50, 11282 Stockholm, Sweden. [3] Department of Medical Biochemistry and Cell Biology, Institute of Biomedicine, University of Gothenburg, Box 440, 405 30 Göteborg, Sweden. ✉email: jonas.a.nilsson@surgery.gu.se

Uveal melanoma (UM) is a rare form of melanoma but the most common intraocular cancer in adults[1]. Enucleation or brachytherapy can provide good local control but in 50% of patients metastases develop, most frequently to the liver and with lethal outcome[2,3]. The genetics of UM has primarily been studied in the primary tumors of the eye, such as in the landscape study by the TCGA consortium[4]. Recurrent mutations in *GNAQ* or *GNA11* are common, whereas mutations in *PLCB4* and *CYSLTR2*, downstream and upstream of *GNAQ/11*, are seen in occasional cases[5–8]. These driver mutations are all largely mutually exclusive. Additional recurrent mutations have been found in *EIF1AX*, *SF3B1,* and *BAP1*, where the latter connotes poor prognosis and development of metastatic disease[9,10]. The development of metastatic UM can also be predicted using gene expression analyses, where Class I transcriptional subtype tumors have an excellent prognosis but Class II is strongly associated with metastasis[11]. However, additional prognostic subgroups can also be determined[4,12–14].

Patients with UM metastases are not predicted to respond to the same targeted therapies as patients with cutaneous melanoma since UM does not have *BRAF* mutations. Moreover, retrospective analyses of outcome following the use of immune checkpoint inhibitors have demonstrated poor response rates at multiple centers[2]. At our center, we are using isolated hepatic perfusion with melphalan to treat patients with liver metastases of UM. During the surgical procedure leading to the perfusion treatment, there are possibilities of procuring fresh biopsies for the generation of PDX models, tumor-infiltrating lymphocyte (TIL) cultures and for genomics studies of metastases (Fig. 1a). Here, we describe a profiling of 32 metastatic UM tumors using whole-genome sequencing and also characterize infiltrating lymphocytes, providing molecular insight into the genomic events and immunology involved in late-stage UM.

## Results

**Recurrently mutated genes in UM metastases**. In total, 32 metastases of UM, 6 subcutaneous, and 26 from the liver (Supplementary Table 1), were collected and subjected to whole-genome sequencing and 28 of them to poly-A+ RNA sequencing. Twenty-eight of the tumors were pathologically designated as originating from the choroid, one in the ciliary body and one in the iris, whereas two cases did not have information about primary uveal location available. All liver metastases came from patients that were untreated at the time of biopsy and 24 of them had been enrolled in the SCANDIUM phase III trial[15]. All cutaneous biopsies except one came from patients previously treated with chemotherapy (IHP, dacarbazine, and/or taxanes).

Variant calling with MuTect 2[16] revealed mutations in *BAP1*, *GNA11*, *GNAQ*, *SF3B1*, *CYSLTR2*, and *PLCB4* (Fig. 1b, Supplementary Fig. 1a, b and Supplementary Data 1), which are recurrently altered in UM[5–9]. We discovered no mutations in *EIF1AX*, which are associated with a good prognosis[9]. In all, 29/32 (91%) of metastases were found to have *BAP1* mutations. These were paired with loss of chromosome 3 in the vast majority of cases (Fig. 1b). In one case, loss of heterozygosity on 3 occurred in a copy number neutral manner (Supplementary Fig. 1c). Notably, *BAP1* was also the subject of alterations not detected by standard variant calling, including one large deletion spanning the first three exons. In another case, an intronic event far from the nearest splice site was associated with novel splicing events and intron retention at the point of the mutation (Fig. 1c). A third tumor contained a 48 bp fully intronic homozygous deletion that again did not occur at a splice site, but associated with missplicing and intron retention clearly tied to the event (Fig. 1d). These two alterations most likely created new intronic splice sites.

A previous study has described a mutation that activates a cryptic splice site within an exon in *BAP1*[17]. To our knowledge, no cases have been described for de novo splice-site-generating intronic mutations in UM; only cases that disrupt canonical splice sites at the exon–intron boundary[18]. As *BAP1* loss predicts metastasis[3], this highlights the need to also investigate intronic non-splice site mutations as candidates for loss-of-function events, which exome or targeted sequencing may not be sufficient to reveal.

Among the three patients where *BAP1* mutations could not be established, two had *SF3B1* mutations. We also detected mutations in *SF3B1* that occurred outside the common hotspots K666 or R625. These included K700E and an in-frame deletion at V577. The first has to our knowledge not been described in UM, but is frequent in other cancer types, including breast cancer[19], chronic lymphocytic leukemia[20], and pancreatic adenocarcinoma[21]. Some *SF3B1* mutations also co-occurred with *BAP1* mutations, illustrating that mutual exclusivity between these events is imperfect[5].

In the third tumor without *BAP1* mutation, we did not discover mutations in either *SF3B1* or *EIF1AX*. *BAP1* nuclear staining was also confirmed with immunohistochemistry (Supplementary Fig. 4d). This tumor (UM28) was also the only one inferred to be tetraploid (Supplementary Fig. 1d, e), and had frequent wide copy number losses, affecting chromosomes 1p, 3, 4q, 6q, 8p, 9, 11, 14, and 16. Mutated genes in these regions included *YEATS2* and *ZMAT3* on chromosome 3 and *AKT1* on chromosome 14. Lack of *BAP1* mutations is characteristic for tumors of the class I subtype, among which only a small subset tends to metastasize. Those that do metastasize have been shown to be distinguishable through elevated *PRAME* expression[13]. This particular tumor displayed the second highest levels of *PRAME* expression (Supplementary Fig. 1f), suggesting that it could have originated from such a class I tumor.

In addition, we found two metastases with mutations in the tumor suppressor *TET2*, in one case leading to a stop-gain. A third tumor had a frame-shift deletion in *TET1*. *TET1* and *TET2* exert epigenetic control via DNA demethylation[22,23]. Some metastases also had mutations in genes that interact with *BAP1*, including *ASXL2* and *FOXK2*[24] (Supplementary Data 1). These mutations were not present in TCGA primary UM samples (Supplementary Fig. 1g, h).

**Mutational signature of UV damage in UM**. The causes that underlie UM are to date largely unknown, and despite risk factors implying a potential role for UV radiation, no clear evidence has emerged to date and the field is divided on whether this can be a driving factor[6,14,25–27]. The pattern of trinucleotide substitutions across the genome can be informative about underlying mutational processes. Therefore, we estimated the relative contributions of established mutational signatures[28] to the total mutational burden in the tumors.

Consistent with previous observations[6,14], the dominating signatures were S1, S3, S5 (COSMIC nomenclature), and to a lesser extent S16. S3 has been associated with defective DNA double-stranded break repair, whereas S1 and S5 are termed clock-like and associate with aging[28,29] (Fig. 1e). However, one tumor had a distinctly different profile, dominated by contributions from S7 (~63%), with a bias toward the untranscribed strand ($q < 0.05$, Poisson test), more closely resembling cutaneous melanomas sequenced concurrently (Fig. 1e, f). S7 is known to arise as a consequence of UV radiation-induced damage[28]. We could exclude a sample mix-up from the presence of the same *GNA11* Q209L mutation and *BAP1* frame-shift deletion in RNA, together with transcriptomic classification against ~10,000 tumors from TCGA (Supplementary Fig. 2a–c).

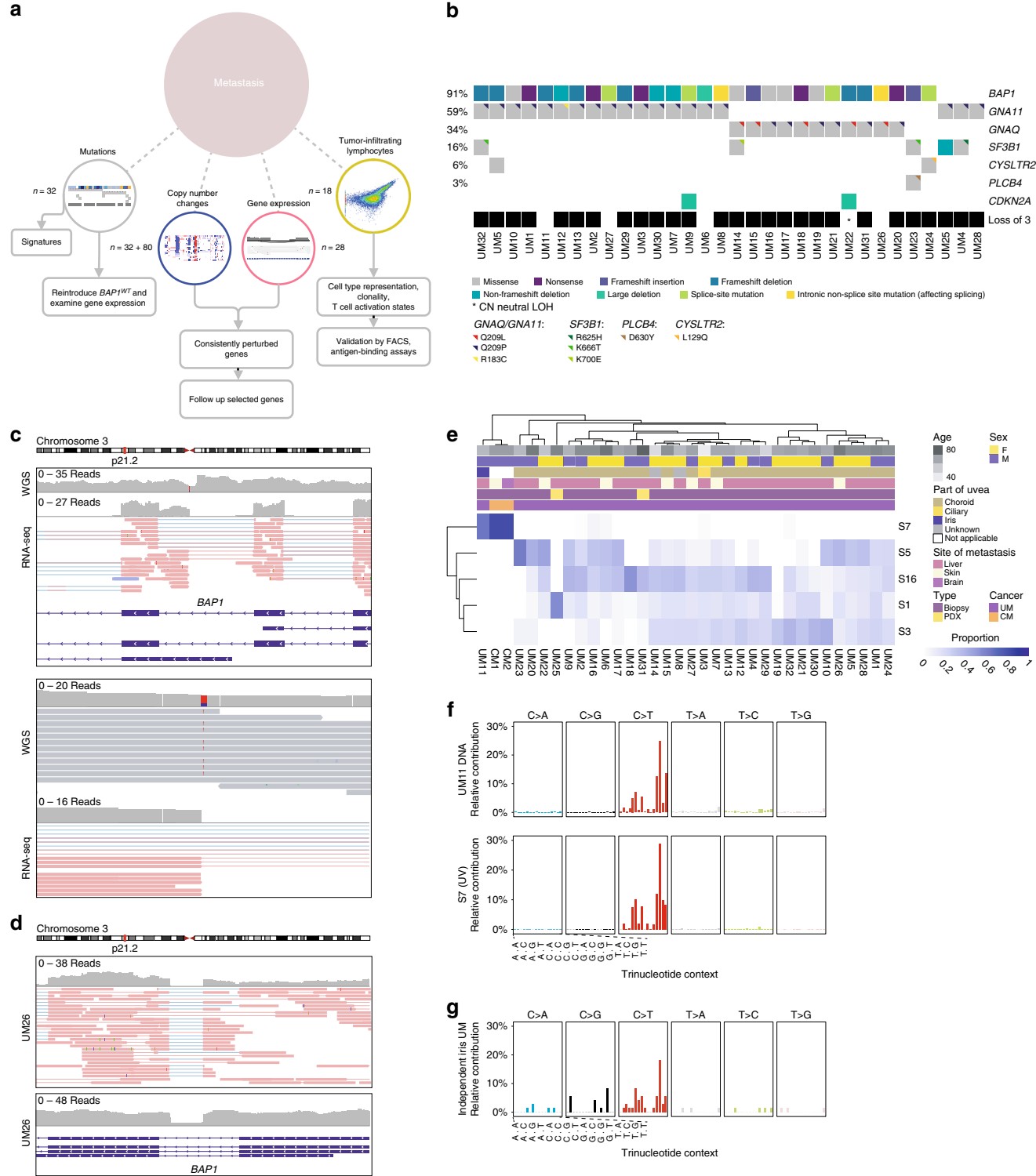

**Fig. 1 Mutations in metastatic uveal melanoma (UM). a** Overview schematic of the study. Thirty-two samples were subjected to whole-genome sequencing and 28 to RNA sequencing. Eighty tumors from TCGA were compared in copy number analyses. TILs from 15 tumors were used for antigen-reactivity assays and 5 of these, as well as 3 other tumors were used for single-cell analyses of TIL phenotypes. **b** Mutations in genes recurrently altered in UM. Chromosome 3 status is indicated. **c** Intronic non-splice site point mutation in *BAP1*, associated with aberrant splicing. **d** Intronic large deletion in *BAP1* associated with aberrant splicing. **e** Estimated contributions of COSMIC mutational signatures. Samples and signatures are ordered by agglomerative hierarchical clustering. Signatures with estimated contribution <30% excluded. $n = 32$ independent UM samples were included and $n = 2$ cutaneous melanomas (representing two metastases from one patient) sequenced at the same time were included for comparison. Signatures were inferred using both synonymous and non-synonymous mutations. CM: cutaneous melanoma. **f** Overall mutational spectrum of UM11, based on WGS data. The canonical profile of the UV-associated "signature 7" is shown for comparison. **g** Mutational spectrum from exome data of an unrelated primary iris melanoma.

We hypothesized that this unexpected signature could be explained owing to the tumor having originated in the iris (Supplementary Fig. 2d–g), a site from which only 3–5% of cases arise[3], compatible with an absence of iris melanomas and UV evidence in the TCGA UM cohort[4]. To confirm this, we managed to obtain a second iris UM sample from a patient without metastasis, which again revealed a prominent UV pattern (Fig. 1g). Thus, although rare, UM can evidently be induced by UV damage if manifest in the iris. Interestingly, this associated with an elevated amount of so called neoantigens, peptide sequences predicted to be immunogenic (Supplementary Fig. 2h).

**Copy number changes overrepresented among metastases.** UM is characterized by highly recurrent copy number aberrations affecting entire chromosome arms[4]. All metastases had gain of chromosome 8q, known to co-occur with monosomy 3 in poor-prognosis tumors[14,30] (Fig. 2a). A number of arm-level changes were also significantly overrepresented in the metastatic tumors compared with tumors studied by TCGA (Fisher's exact test, $q < 0.05$). These were loss of 17p, 6q, and chromosome 3, as well as gain of 8q and 5p (Fig. 2a, b, Supplementary Data 2). Previous studies have also found loss of 6q and 8p to be overrepresented in metastatic tumors[14,30]. Sequencing of matched primary tumors for UM16 and UM24 showed that gain of 5p in UM16 and loss of 6q in UM24 were late events only present in the respective metastases, whereas 8q gain and loss of 3 was present already in the primaries in both cases (Fig. 2c). Loss of 6q has previously been found to associate with metastasis[31]. Overall, genomic losses tended to be more frequent in these metastases than observed in TCGA tumors.

Focal events were very rare. Notably, however, we discovered somatic focal deletions affecting CDKN2A and the nearby gene MTAP in two samples (Fig. 2d, Supplementary Fig. 3a, b). CDKN2A encodes the tumor suppressors p16[INK4a] and p14[ARF] and is commonly deleted in cutaneous melanoma[32]. The deletions here were homozygous and hemizygous respectively. While CDKN2A expression was still present in the hemizygous case, a subsequent patient-derived xenograft (PDX) model established from this tumor (Supplementary Fig. 4) showed full loss of expression, even extending to other nearby genes (Fig. 2e, Supplementary Fig. 3a). This suggests that either a pre-existing clone with a homozygous deletion or a second loss event was selected for as the tumor established itself in this new environment, supporting CDKN2A loss as a late event that may be relevant in the metastatic setting[33].

**Gene expression associated with recurrent copy number events.** To understand how the recurrent chromosomal events in UM affect the transcriptome and to rank genes by a potential to influence tumor behavior, we searched for consistent correlations between the copy number and gene expression in both this data set and TCGA UM, and ordered them by their degree of known protein–protein interactions from the Human Protein Reference Database (HPRD), followed by association with observed survival. The top candidates per region are shown in (Fig. 2f, Supplementary Data 3). An analysis using the "chemical and genetic perturbations" collection in MSigDB showed that regions of gain were enriched for the category genes upregulated in class II UM ($q < 6.11 \times 10^{-14}$), whereas regions of loss were enriched for genes downregulated in class II[34] ($q < 5.49 \times 10^{-11}$). The class II transcriptional subtype is one of the two major subdivisions of UM, strongly associated with metastasis[34]. Pathway enrichment analysis revealed processes that included transcriptional regulation, stress responses, immune signaling, and developmental biology (Fig. 2g, Supplementary Data 3).

Top ranked genes in loss regions included CASP9, an early activator of apoptosis[35] and the aforementioned CDKN2A. Candidates in gain regions included MAPK14 (p38α), a kinase that operates at the intersection of cell cycle progression, stress signaling, immune responses, and differentiation[36], and the very recently proposed UM oncogene PTK2 (FAK)[37], a focal adhesion associated kinase known for being activated upon matrix–integrin interactions and thereby mediating survival signals that prevent detachment-associated apoptosis (anoikis)[38,39]. A small RNAi screen in three cell lines, directed against a list of genes selected based on gain candidates, in a cell line derived from the UM22 tumor demonstrated that a majority of siRNA pools negatively affected proliferation (cell count) or viability (ATP production) to a similar or higher level than an siRNA against GNAQ (Fig. 2h, Supplementary Fig. 3c, d, Supplementary Data 3). Thus, these recurrent arm-level copy number changes contribute to shaping the transcriptomic subtypes of UM and regulate genes that may conceivably contribute a fitness advantage.

**Reversal of transcriptomic subtype upon BAP1 reintroduction.** We next asked to what extent BAP1 mutations could influence the transcriptome of metastatic UM. For this purpose, we used the UM22 cell line, which had been established from one of the metastases grown as a PDX, and which had a homozygous frame-shift deletion in BAP1, but no copy number loss on chromosome 3 (Supplementary Fig. 1c, Supplementary Fig. 4, and Supplementary Fig. 5a). Mutations do not always cause a complete loss of the BAP1 protein and in UM22 the mutation generated a translation stop before the nuclear localization signal. Therefore, the protein resided predominantly in the cytoplasm, as opposed to the nucleus, where it is commonly found in BAP1 wild-type tumors (Supplementary Fig. 4b, c). A functional copy of BAP1 was introduced using a retroviral vector and RNA-seq performed on this and an empty vector control sample (Fig. 3a). Immuno-histochemistry (IHC) and RNA-seq alignments showed successful integration and expression of the wild-type BAP1 allele (Fig. 3b, c, Supplementary Fig. 5b–e). A differential expression analysis between the two conditions revealed a large transcriptomic response, with 518 genes downregulated and 990 upregulated at an absolute $\log_2$ fold change $>1$ and $q < 0.05$ (6707 in total without fold change cutoff, Fig. 3d, Supplementary Data 4). SLC7A11, identified by Zhang et al.[40] as a mediator of ferroptosis-suppressive effects of BAP1, was significant albeit not as strongly regulated ($\log_2$ fold change $= -0.82$, $q = 5.42 \times 10^{-19}$). Pathways enriched among downregulated genes upon reintroduction included GPCR signaling, neurotransmitter receptor transmission, interferon α/β signaling and chemokine activity. Upregulated pathways most prominently included post-transcriptional and translational mechanisms (Fig. 3e).

Notably, we observed significant regulation of 9 out of 12 genes used as discriminating features in a classifier that distinguishes between the high-risk class II versus class I subtypes[41], some of which are melanocyte lineage markers and a few of which have also been found compatibly regulated upon silencing[10] (Fig. 3d). These genes were all expressed in the inverse fashion expected for class II tumors, with CDH1, ECM1, and HTR2B decreasing upon BAP1 reintroduction and LMCD1, LTA4H, MTUS1, ROBO1, SATB1, and FXR1 increasing, whereas the remaining three genes, RAB31, ID2, and EIF1B, were not significant. RT-qPCR measurements confirmed the RNA sequencing data for all genes but FXR1 (Fig. 3f).

To investigate whether the apparent transcriptomic shift towards the class I subtype was limited to these few discriminating genes or representative of a broader change, we performed a gene set enrichment analysis on the whole list of differentially

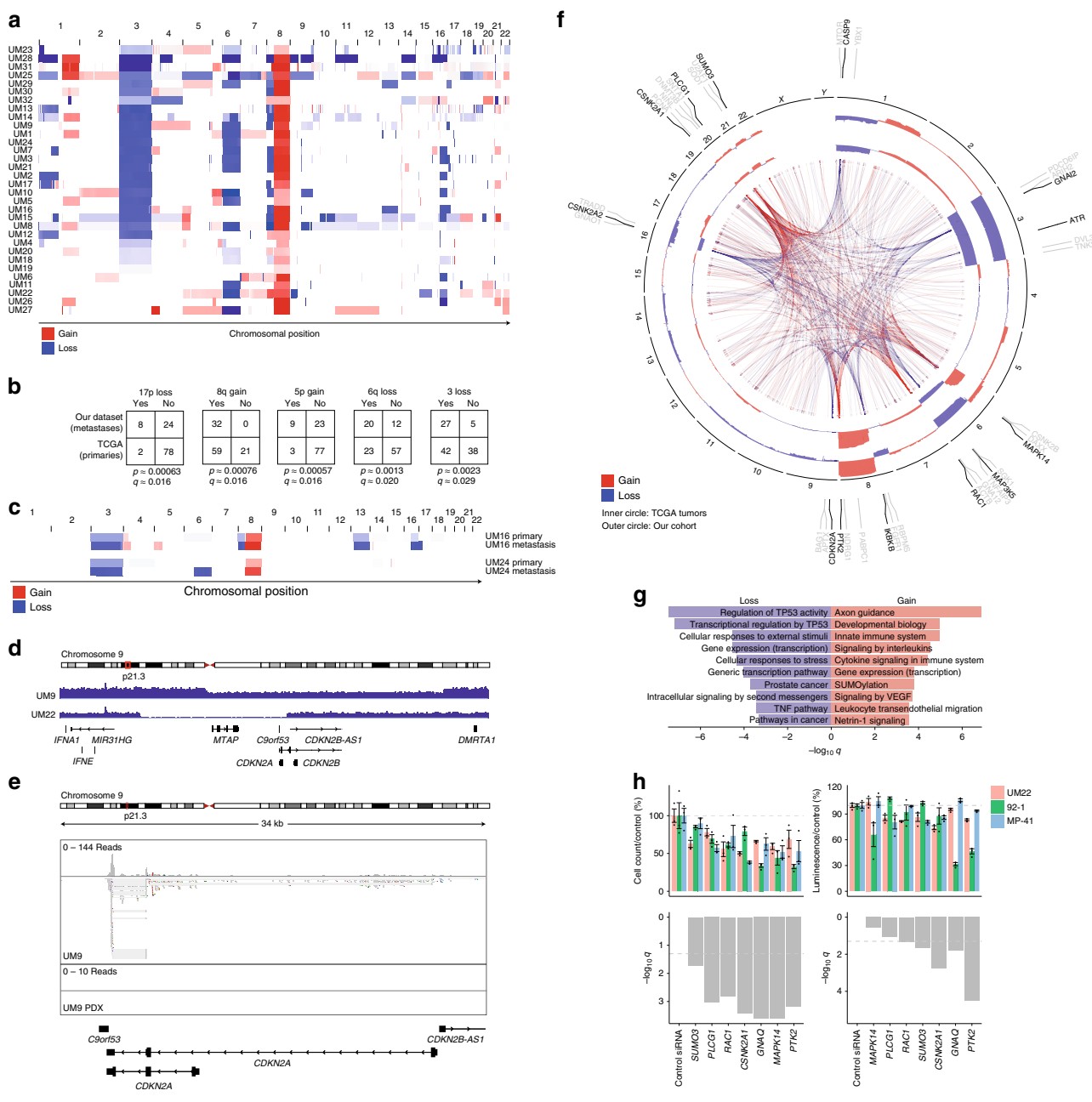

**Fig. 2 Copy number analysis. a** Copy number profiles of each tumor. Differences in color intensity depend on copy number amplitude and tumor purity. **b** Broad copy number changes enriched in the metastases ($n = 32$) compared with TCGA tumors ($n = 80$). Two-tailed Fisher's exact tests with adjustment for multiple testing using the Benjamini–Hochberg method. **c** Two primary tumors compared with matched metastases. **d** Focal deletions of *CDKN2A* in two samples. **e** RNA-seq from the UM9 metastasis and corresponding PDX showing the region with focal *CDKN2A* deletion. **f** Genes in recurrent arm-level copy number aberrations ranked by associations between gene expression and copy number that were consistent in this cohort and TCGA tumors, and further ranked by protein–protein interaction network degree from the Human Protein Reference Database (HPRD), and additionally by presence of any associations with worse survival. The top three candidates are shown in each region. Connecting lines represent protein interactions of the highest ranked gene per region. Blue represents regions of loss and red regions of gain. Summarized representations of copy number profiles per region show the relative numbers of gain and loss events, with the inner circle representing TCGA samples and the outer our cohort. For the metastatic cohort, $n = 28$ samples with matching DNA and RNA data were included. **g** Pathways enriched among the combined set of the top 10 genes per region of gain or loss. **h** Functional interrogation by siRNA of main candidate genes whose expression is elevated due to copy number alteration. Cells were counted or viability was measured at 72 h, 96 h and 96 h for the cell lines UM22, MP-41, and 92-1, respectively, after transfection of the siRNA pools. $n = 3$ samples were transfected independently, for each cell line. Data are presented as mean values ± standard error of the mean (SEM). Two-way ANOVA was used to estimate differences, taking into account both cell line and target gene as variables. $q$ values were calculated using Benjamini–Hochberg correction, taking into account all genes in **h** as well as those in Supplementary Fig. 3c, which shows other candidates of interest investigated. Dotted lines indicate $q = 0.05$.

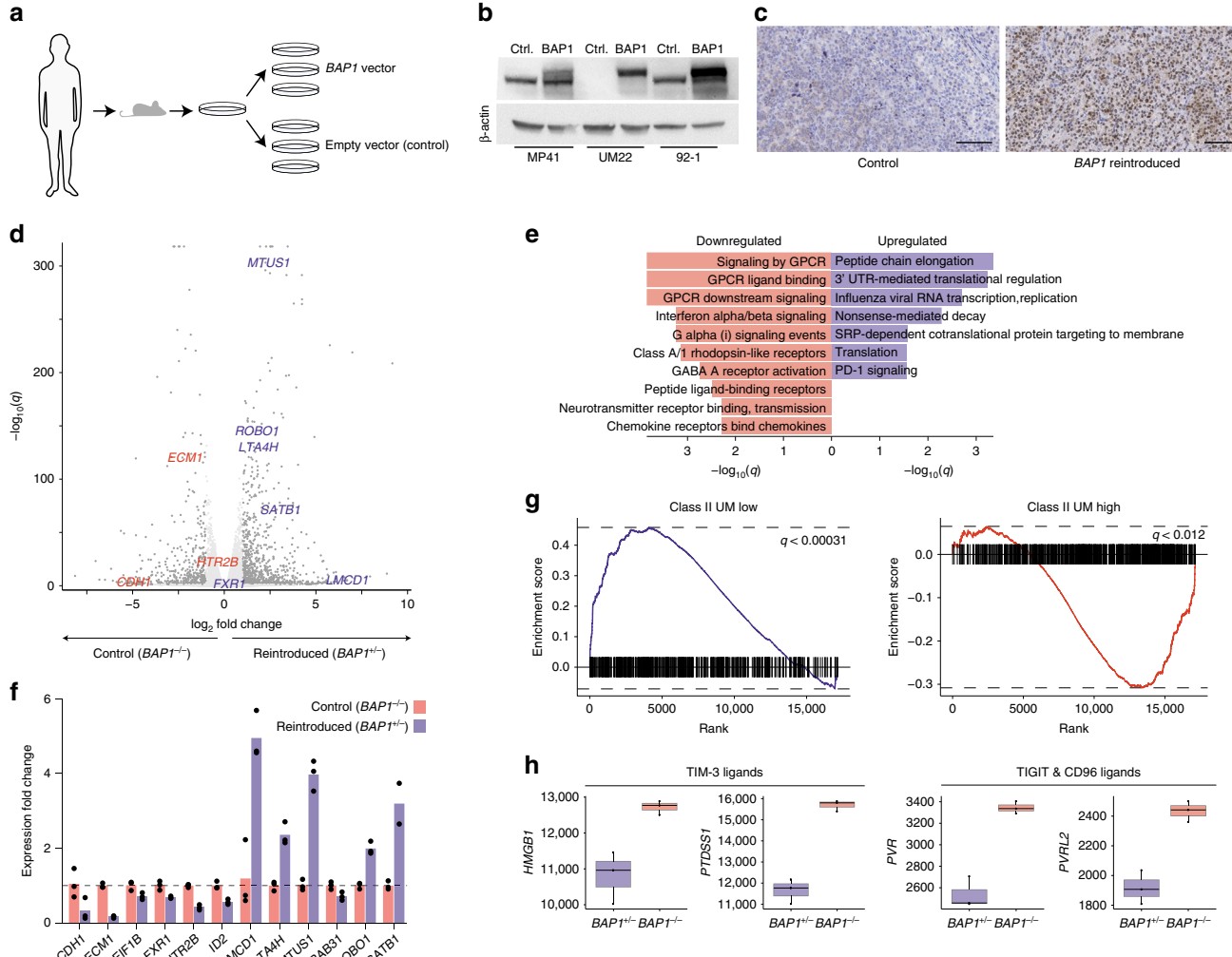

**Fig. 3 Reintroduction of *BAP1* into a deficient tumor. a** Schematic representation of the experiment. Cell lines from a PDX model established from tumor UM22 were transduced with either *BAP1* wild-type containing viral vectors or empty vectors and subjected to RNA sequencing. **b** BAP1 protein levels in empty vector controls and *BAP1*-reintroduced cells. The *BAP1* wild-type cell lines MP-41 and 92-1 are shown for comparison of expressed BAP1 levels with the same respective treatments. Full gel images are shown in Supplementary Fig. 5e. The western blot was repeated twice with similar results using biological replicates. Ctrl: control. **c** IHC staining in PDX models for BAP1 expression in empty vector controls and *BAP1*-reintroduced cells. Scale bars represent 100 μm. Shown is one staining per cell line performed on sections from tumors from one mouse out of three transplanted mice per cell line. **d** Differentially expressed genes for $q < 0.05$ and absolute $\log_2$ fold change > 1. $n = 3$ independently grown samples of cells derived from either the case or control cell lines, respectively, were used and differences were assessed using DESeq2. Genes from a clinical assay distinguishing the class I and II UM subtypes are indicated[41], with blue indicating upregulation in *BAP1*-reintroduced cells, and red representing downregulation. **e** Top 10 enriched Reactome gene sets. **f** RT-qPCR results for the genes indicated in **d**, with $n = 3$ technical replicates. **g** Gene set enrichment analysis with respect to the MSigDB chemical and genetic perturbations category, with results from the two sets discriminating between class I and II subtypes shown[34,42]. **h** Downregulation of immune checkpoint ligand-related genes upon *BAP1* reintroduction, as determined from RNA-seq data on $n = 3$ independently grown samples of cells derived from either the case or control cell lines, respectively. Horizontal lines indicate median, lower, and upper bounds of boxes represent the first and third quartiles, whiskers represent the smallest/largest data point at most 1.5 times inter-quartile range from the lower and upper bound, respectively.

expressed genes using the "chemical and genetic perturbations" collection from MSigDB[42]. We found that a signature of genes lowly expressed in class II UM[34] was significantly enriched among upregulated genes ($q < 0.00031$), and that genes highly expressed in class II were enriched among downregulated genes ($q < 0.012$), showing that this trend is indeed broader (Fig. 3g, Supplementary Data 5). This shift towards the class I subtype upon *BAP1* reintroduction implies that the inverse drives the cells towards the metastatic class II transcriptional subtype, which characteristically has *BAP1* alterations. We performed proteomics by mass spectrometry analysis of three replicates from both conditions, and found this theme to be similarly recapitulated among differentially expressed proteins (Supplementary Fig. 5f–h,

Supplementary Data 6). Moreover, 2358 of the genes ($q < 0.05$) also differed between *BAP1* mutated and wild-type TCGA tumors in a way that corresponded to changes observed when reintroducing the gene (Supplementary Fig. 5i–l), further illustrating that genes relevant for in vivo differences between the subtypes are altered in the experiment.

Beyond this, *BAP1* restoration also downregulated the TIM-3 immune checkpoint ligand-related genes *HMGB1* and *PTDSS1* as well as the *TIGIT* and *CD96* ligands *PVR* and *PVRL2*, implying higher expression levels in *BAP1*-deficient UM cells (Fig. 3h, Supplementary Data 4). Mass spectrometry protein measurements confirmed this for all except *PVRL2* (Supplementary Fig. 5h). Similarly, in TCGA UM samples, all

except *PVRL2* were expressed in a compatible manner (Supplementary Fig. 5l).

**Tumor-reactive T cells present in UM metastases.** Having observed regulation of immune-related pathways in UM cells, we next investigated the phenotypes of TILs isolated from metastases. We performed flow cytometry of cryopreserved single-cell preparations of tumors and found that fractions of CD8$^+$ and CD4$^+$ T cells differed between biopsies from different patients (Fig. 4a). Expression of PD-1 and CD39 have been proposed to mark tumor-reactive T cells as opposed to bystander tissue resident cells[43]. High fractions of PD-1$^+$CD39$^+$CD8$^+$ T cells were found in a subset of samples (Fig. 4a, gating strategy in Supplementary Fig. 6a, b). Antigen specificity can be detected by flow cytometry using tumor-associated antigen peptide:HLA dextramers. Most of the obtained biopsies contained too few cells to enable detection of many different melanoma-associated antigens. However, upon receipt of the biopsies we had also expanded so called young TILs (yTILs) from many samples in IL-2 containing media, e.g., for use in studies of adoptive T-cell transfer in melanoma PDX models[44,45]. Eight out of fifteen HLA-A2:01-positive patients' yTIL cultures had MART-1 dextramer or gp100 dextramer positive cells (Fig. 4b, Supplementary Fig. 6d, Supplementary Data 7). To gain insight into the heterogeneity of clonotypes of TCRs we profiled eight yTIL cultures by single-cell RNA and TCR sequencing. In all eight yTIL cultures single-cell sequenced we observed CD8$^+$ T cell clones that expressed *PDCD1* (PD-1) and *ENTPD1* (CD39) (Fig. 4c). Expression-based clusters were formed both by cell type and T cell receptor clonotype (Supplementary Fig. 7a–c), the latter partially explained by different activation states (Supplementary Fig. 7d, e, Supplementary Fig. 8). Besides PD-1 and CD39 (Fig. 4c, Supplementary Fig. 6c, Supplementary Fig. 7d, e), other inhibitory receptors were also prominently expressed in both biopsies and yTILs: TIM-3 was high in both, whereas TIGIT was higher in yTILs than in biopsies (Fig. 4d, Supplementary Figs. 8–10). Moreover, LAG3 was also expressed in some yTILs (Supplementary Fig. 8b). Collectively our data suggest presence of tumor-reactive TILs in UM tumors, large intra-patient heterogeneity and expression of immune checkpoint proteins. Expression of TIM-3 and, to a lesser extent TIGIT, and potentially ligands related to these, could indicate means of immune evasion in UM that are different from cutaneous melanoma.

## Discussion

Metastatic UM currently entails a very poor outcome owing to the lack of effective treatment options[3]. Genetics of the primary disease confined to the eye has already been investigated in several hallmark studies[4–6,9,10]. However, only a few metastatic samples have been sequenced with exome or whole-genome sequencing and our study has the largest sample cohort sequenced to date with whole-genome sequencing. A history of primary UM and lack of therapeutic efficacy of surgery makes biopsy and surgical removal of metastatic samples uncommon. By obtaining biopsies of liver metastases from patients in the SCANDIUM trial or cutaneous metastases of UM, we have been uniquely positioned to focus on the metastatic disease by both analyzing fresh frozen material by genomics as well as generating PDXes, cell lines, and TIL cultures for transcriptomics analyses.

The key event to metastasis in UM is loss of the tumor suppressor *BAP1*[10]. Compatible with this, we observed *BAP1* mutations in 91% of the metastases. Two of these seem to have altered splicing via intronic events outside of canonical splice regions, via creation of new intronic splice sites. This illustrates special cases that exome sequencing may not be sufficient for detecting. Given the implications of *BAP1* status, one may therefore argue for more comprehensive sequencing at this locus.

It has been reported that mutations in the epigenetic regulators *PBRM1* and *EZH2* can occur late during metastatic development of UM[33]. Consistent with those observations, we also find a mutation in *EZH2* in one of the samples. In addition, we find mutations in other not previously implicated epigenetic regulators, including *TET1*, *TET2*, and *ASXL2*, the latter of which is known to interact with *BAP1*[22–24]. This may support that such alterations can be relevant to additional selection pressures a tumor may be subjected to during metastasis.

We furthermore find that out of two tumors studied of the iris subtype, both had mutational spectra associated with UV-induced damage. Mutational signatures of UV damage in UM have not previously been reported and a consensus of UV-involvement in UM has not been reached by previous epidemiological studies. Although iris UM is rare, the metastasis studied here had much higher than average mutation load, and predicted number of neoantigens. This could potentially render such tumors suitable for immunotherapy, which otherwise lacks efficacy in UM. Interestingly, the iris UM metastasis concerned here also harbored T cells recognizing MART-1.

Several broad copy number events were found to be more frequent in the metastases studied compared with primary tumors from TCGA, including losses of chromosome 3, 6q, and 17p, as well as gains of 5p and 8q. Notably, 8q gain was present in every metastasis. By sequencing matched primary tumors for two cases, we could establish that in one of the tumors 5p gain and 17p loss had arisen during metastasis, and in the other case 6q loss. Furthermore, two tumors had focal deletions of *CDKN2A*, an event that may have a larger relevance in the metastatic setting, as it has not been detected in recent large-scale studies of primary UM tumors[4–6,14,46] and since it has also been observed to be deleted in a step-wise fashion during metastatic progression in UM[33]. The latter is consistent with our finding that a PDX model established from a tumor with a hemizygous deletion was revealed to have homozygous loss of *CDKN2A*, suggesting a selective advantage for a secondary inactivating event.

We additionally mapped out genes with correlations between expression and arm-level copy number changes in both this data set and that of TCGA and ranked them by their degree of protein–protein interactions and any associations with survival present to gain an understanding for central processes affected and potential targets. We found several interesting candidates, including the recently proposed UM oncogene *PTK2*[37], *MAPK14*, the apoptosis mediator *CASP9*[35], as well as *CDKN2A* to be first-ranked candidates in 8q gain, 6p gain, 1p loss, and 9p loss, respectively. We performed a siRNA knockdown experiment against selected genes and found proliferation and viability decreases to be the consequence when targeting the majority of those. In addition, we found expression changes mediated by loss events to be enriched for genes generally downregulated in poor-prognosis tumors and gain events enriched for genes upregulated in poor-prognosis tumors, showing that these broad events contribute to shaping the distinct transcriptomes of the two subtypes.

To increase our understanding for how these transcriptomic subtypes are established, we investigated the contribution from *BAP1* loss by reintroducing a functional allele into a cell line established from one of these metastases. This cell line is unusual in the sense that it proliferates well in culture, whereas BAP1-deficient UM cell lines generally are associated with extremely slow growth characteristics[47]. It is, in fact, the only BAP1-deficient UM cell line we have been able to grow well enough to study. It is tempting to speculate that the genetic make-up of this cell line, with a diploid chromosome 3 and a

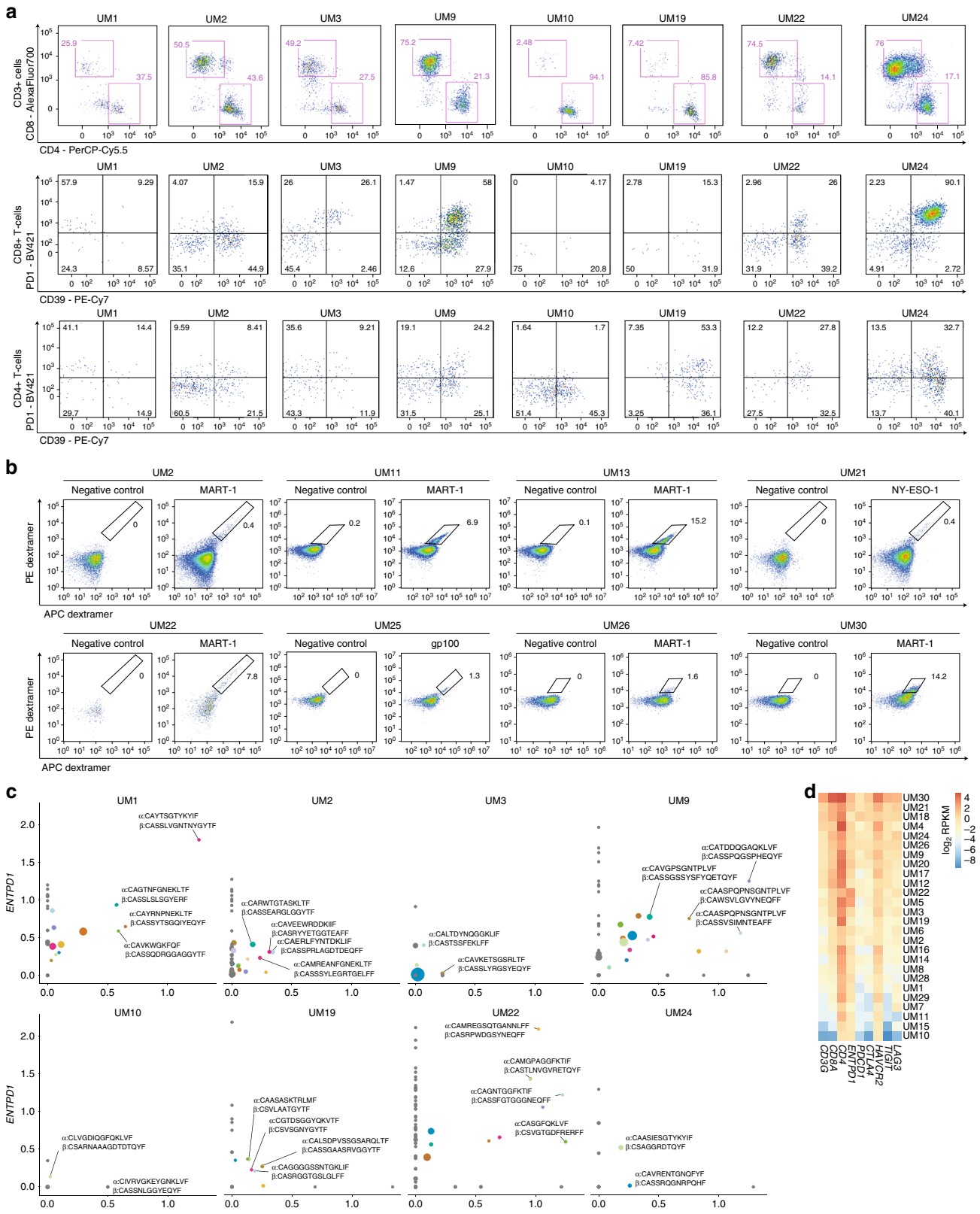

downstream mutation of *BAP1* that impairs the nuclear localization signal, but leaves the remainder of the protein unaffected, is the reason for this favorable growth in culture. Nevertheless, we show in this one, and potentially unique, cell line that reintroduction of a wild-type *BAP1* resulted in a reversal of the transcriptomic subtype from class II to class I, which was also seen at the protein level. The transcriptomic changes are in accordance with the phenotypic switch observed upon genetic inhibition of *BAP1* in *BAP1* wild-type cells[48]. Of potential relevance was also the down-regulation of TIM-3 and TIGIT ligand-related genes after restoring *BAP1* function, indicating potential upregulation on

**Fig. 4 Analysis of tumor-infiltrating lymphocytes. a** Proportions of CD8[+] and CD4[+] T cells from biopsy material, and proportions of these that were positive for PD-1 and CD39. Sample UM22 was derived from a patient that has previously been treated with chemotherapy, possibly affecting TIL proportions in this sample. **b** Assessment of T-cell reactivity against MART-1, gp100 and NY-ESO-1 in yTIL cultures. Proportions found to be reactive are indicated. Samples tested were those with the HLA-A*02:01 genotype, as this genotype is known to present MART-1 and gp100. Samples with this genotype (Supplementary Data 7) that are not shown were also tested and found to be negative. **c** Analysis of relative levels of *PDCD1* (PD-1) and *ENTPD1* (CD39) expression among different CD8[+] T-cell clonotypes, determined by single-cell RNA-seq of yTILs. Clonotypes with one pair of alpha and beta chain were included, and the ones with greatest expression of both markers are highlighted. Point sizes are proportional to clonotype frequency. Gray color indicates clones that were negative for either *PDCD1* or *ENTPD1*, whereas other colors indicate different clonotypes that correspond to those in Supplementary Fig. 7e. **d** Expression of T-cell markers and checkpoint receptors in bulk RNA-seq data from biopsies (batch-corrected log$_2$ RPKM normalized values). *yTILs* young TILs, TILs isolated and expanded from a biopsy with a low dose of IL-2.

*BAP1* loss that may have consequences for tumor–immune interactions.

We profiled tumor-infiltrating lymphocytes isolated from some of the metastases by flow cytometry and found tumor-reactive subsets present in several cases. This argues that lack of immune responses is not due to bystander T cells being the only ones present in the tumors. However, we also noted expression of the checkpoint receptors PD-1, TIM-3, LAG3, and, to some extent, TIGIT. Potentially, the inferred upregulation of ligands for TIM-3 and TIGIT upon *BAP*1 loss may cooperate with the expression of these receptors on T cells to interfere with antitumor immunity. Given the historic failures of anti-PD-1 and anti-CTLA-4 therapies in UM, this may argue for exploring these other checkpoint receptors. After publishing our preprint, another study performing scRNA-seq of TILs in UM biopsies found results that support this[49]. Overall, however, our molecular profiling of TILs suggests large intra-patient heterogeneity of both TILs residing in the biopsies as well as in expanded TILs. This heterogeneity could contribute to infrequent or non-durable tumor responses to checkpoint inhibitors as well as adoptive T-cell transfer of TILs[3].

Collectively, these results highlight that exome sequencing might not be sufficient for detecting *BAP1* loss, which is the most significant event in UM metastasis, that UV damage can be an important mutational process in the iris subtype and that recurrent copy number aberrations cooperate with *BAP1* loss to shape the transcriptome of the metastatic subtype. We also describe immune-profiles of T cells present in metastases that indicate tumor recognition, and heterogeneous activation states of T-cell clones, some with expression of checkpoint receptors that are not targeted by current immunotherapies.

## Methods

**Processing of tumor biopsies**. The patients received oral and written information and signed informed consent agreements according to the ethical approval at the Regional ethical review board (#289-12 and #144-13). Biopsies were either extracted from subcutaneous metastases or from liver metastases, during the procedure of isolated hepatic perfusion in the SCANDIUM trial (NCT01785316) for those participating in it. Two cutaneous melanoma tumors used for comparison were derived from a subcutaneous and brain metastasis of a previously profiled case[50]. Tumor biopsies were divided into pieces that were snap-frozen or minced and used for cryopreservation or tumor-infiltrating lymphocyte cultures. Snap-frozen tumor pieces were homogenized using Bullet Blender (Next Advance, Troy, NY). DNA and RNA were extracted using the AllPrep DNA/RNA kit (Qiagen, Hilden, Germany). Primary eye tumors were formalin-fixed and paraffin embedded (FFPE) in blocks at St Erik's Eye Hospital's pathology biobank.

**Sequencing**. DNA and RNA from fresh frozen biopsies, blood and tumor-infiltrating lymphocytes were extracted as described. Primary eye tumors were sectioned and processed using an FFPE DNA kit (Qiagen). Whole-genome DNA libraries were made using the Illumina TruSeq PCR-free kit or Illumina TruSeq Nano in some cases where low input material was available and sequenced with Illumina HiSeq X Ten at SciLifeLab in Stockholm or Illumina NovaSeq 6000 at GeneCore SU in Gothenburg. RNA libraries were made using the Illumina TruSeq Stranded mRNA kit with poly-A selection and sequenced with Illumina HiSeq 2500 at SciLifeLab in Stockholm or with Illumina NextSeq 500 at GeneCore SU in Gothenburg. Exome-sequencing libraries were prepared with the NextSeq 500 HighOutput Kit v2 and sequenced with Illumina NextSeq 500.

**Preprocessing of DNA sequencing data**. DNA sequencing reads were aligned to the 1000 Genomes version of the GRCh37 reference genome with bwa[51] (v. 0.7.12; options "mem" and "-M"). Duplicates were marked with Picard (version 1.109; https://broadinstitute.github.io/picard). The resulting BAM files were recalibrated with GATK BaseRecalibrator (v. 3.3.0)[52], using known polymorphic sites from dbSNP v138 and 1000 Genomes. PDX samples were aligned separately to the human reference genome and to the GRCm38 version of the mouse reference genome. Reads originating from human were then distinguished using Disambiguate (v. 2018.05.03)[53] with the parameter "-a bwa".

**Preprocessing of RNA sequencing data**. RNA sequencing reads were aligned to the 1000 Genomes version of the GRCh37 reference genome with STAR[54] (v. 2.7.1a) with the parameters "–twopassMode Basic –outFilterType BySJout, –outSAMmapqUnique 60". Splice junctions were provided from the Ensembl GRCh37.75 reference annotation. Gene expression was quantified using htseq-count[55] (v. 0.6.0), with parameters "-m intersection-strict -s reverse". PDX samples were aligned separately to the human reference genome and to the GRCm38 version of the mouse reference genome with STAR. Reads originating from human were then distinguished using Disambiguate, with the parameter "-a star".

**Variant calling**. Variant calling was performed with MuTect 2 (GATK v. 4.0.11.0) using the 1000 Genomes version of the GRCh37 reference genome and a panel of normals. Matched normal samples were used when available. Known population variants were provided from the Genome Aggregation Database (gnomAD). Parameters used were:, "–af-of-alleles-not-in-resource 0.0000025 –disable-read-filter MateOnSameContigOrNoMappedMateReadFilter –genotype-germline-sites true –genotype-pon-sites true –all-site-pls true". The panel of normals was constructed by first running MuTect 2 on each normal with the parameter "–disable-read-filter MateOnSameContigOrNoMappedMateReadFilter" and then merging the resulting lists with GATK CreateSomaticPanelOfNormals. MuTect 2 calls were classified according to quality with GATK FilterMutectCalls and variants failing those filters were removed. Exceptions were made for known hotspot mutation sites in *GNA11*, *GNAQ*, *SF3B1*, *PLCB4*, *CYSLTR2*, and *EIF1AX*. Low-quality variants in *BAP1* were inspected further for support on DNA and RNA alignments. Variant annotation was performed with VEP (v. 91.3) and ANNOVAR[56] (v. 2016-05-11), using the databases COSMIC (v. 79), ESP6500 ("siv2_all"), 1000 Genomes ("2015aug_all"), and dbSNP ("snp138NonFlagged"). For two of the tumors, exome-sequenced normals were used for further filtering using GATK SelectVariants. Comparisons with TCGA UM mutations were made against mutation lists downloaded from GDC (accessed May 29, 2017).

**Mutational signature analysis**. To determine mutation spectra, all somatic autosomal mutations (including synonymous) not present in any population variant resource, and with minor allele read support ≥10, were converted into a 96-trinucleotide mutation frequency matrix using the function "mut_matrix" from the R package MutationalPatterns[57] (v. 1.10.0), with the parameter "ref_genome = 'Bsgenome.Hsapiens.UCSC.hg19'". Known mutational signature trinucleotide frequencies, obtained via COSMIC (http://cancer.sanger.ac.uk/cancergenome/assets/signatures_probabilities.txt; accessed October 27, 2017), were then fitted to the observed mutations using the function "fit_to_signatures" from the same R package. This algorithm operates by searching for the non-negative linear combination of the predefined mutational signatures that best explains all mutations in a given sample, which is done by solving a non-negative least squares optimization problem[57]. This results in estimates of the relative contributions of known mutational signatures in each sample.

**Pan-cancer transcriptomic classification**. RNA sequencing data for 9,583 tumors from 32 cancer types were downloaded from the cgHub repository on December 18, 2015 and aligned to the hg19 human genome assembly, excluding alternative haplotype regions, with hisat[58] 0.1.6-beta (parameters: "–no-mixed –no-discordant –no-unal –known-splicesite-infile"), using splice junctions defined in the GENCODE (v. 19) reference human genome annotation. Gene read counts were derived with htseq-count[55] (parameters: "-m intersection-strict -s no"). RPKM normalized

values were calculated, taking into account the max mature transcript length of each gene and using robust size factors calculated using the median ratio method[59]. For the correlation analysis, reads from our sample of interest were realigned and read counts requantified and normalized using the same methods described for TCGA data. However, standard read depth-based size factors were used for the RPKM normalization of this sample. Pairwise Spearman correlation coefficients were then calculated between our sample and each TCGA sample, with respect to all coding genes (using the function "corr" in MATLAB R2018a). For t-distributed stochastic neighbor embedding (t-SNE) analysis, $\log_2$ transformed (pseudocount of 1 added) expression values of all coding genes were used as input to the "Rtsne" function from the R package of the same name[60]. A separate classification was performed using a $k$-nearest neighbor approach based on Spearman correlations, using $k = 6$, as previously found to be optimal based on leave-one-out cross-validation on the TCGA cohort[50]. With this approach, any ties are broken by taking the majority vote after removing the worst correlating sample.

**HLA genotyping and neoantigen prediction.** HLA genotyping was performed using polysolver (v. 1.0)[61] on whole-genome sequencing data, with the parameters "Unknown 0 hg19 STDFQ", and on RNA with OptiType (v. 1.3.2, default parameters)[62]. For neoantigen prediction, mutated 17-mer peptide sequences centered at each mutation were constructed from non-synonymous point mutations not present in any population variant resource. Predictions against the HLA class I genotypes of each sample were then performed using netMHCpan[63] (v. 4.0, default parameters), considering only 9-mers. Peptides with predicted affinity <500 nM were retained and those deriving from transcripts without expression were removed. Transcript-level expression was quantified using kallisto[64] (v. 0.44.0, default parameters), based on cDNA sequences corresponding to the Ensembl annotation of the GRCh37 human reference genome.

**Copy number segmentation and purity estimation.** Copy number segmentation was performed using binocular (https://sourceforge.net/projects/binocular), using unfiltered variant calls as input, together with aligned reads for tumor and normal samples. Parameters used were "–delta = 90 –min-maf-delta = 0.05 –ai-cutoff = 0.001 –min-copy-ratio = 1.1" for the majority of samples, although for samples with more variable coverage this threshold was raised. Re-centering of segmentation values was required for some samples where estimated values globally deviated from diploid and additional amplitude-based filtering was performed for samples with noisy segmentation profiles resulting from variable coverage or tumor and normals being sequenced on different platforms. For tumors without matching normals, the intersect of segments defined using normals from other samples were used. Sample purity and ploidy was estimated with ichorCNA[65], using the parameters "–ploidy "c(2,3,4)" –normal "c(0.1,0.2,0.3,0.4,0.5,0.6,0.7,0.8,0.9)" –maxCN 10" with tumor BAM files as input. Copy number neutral loss of heterozygosity was determined using the CNVkit (v. 0.9.6a0)[66] "scatter" command, with variants concordant with matching normal DNA as input.

**Associations between copy number changes and metastasis.** Segmented copy number data from TCGA primary tumors were downloaded from GDC Data Portal (accessed 6 October 2017). Copy number changes with an absolute $\log_2$ ratio relative to diploid chromosomes <0.2 and with width less than half the size of the shortest chromosome arm were filtered out from both TCGA UMs and our tumors. The general regions to test were defined as those where a contiguous altered region spanning all events in all metastasis samples could be constructed that had the required width and amplitude and which was present at least once in either dataset. Changes of the same direction (loss or gain) affecting each region were then assessed for association with each of the two data sets using Fisher's exact test (two-tailed) using the function "fisher.test" in R 3.6.0. As the resulting contiguous regions practically spanned the entire length of each affected chromosome arm, with the exception of chromosome 3, which spanned the entire chromosome, one test was performed per gain or loss event of each such arm. $p$ values were corrected for multiple testing using the Benjamini–Hochberg method.

**Ranking of genes in broad copy number aberrations.** RNA-seq data for the TCGA-UVM data set ($n = 80$) were downloaded using TCGAbiolinks[67], with parameters "project = 'TCGA-UVM', data.category = 'Transcriptome Profiling', data.type = 'Gene Expression Quantification', workflow.type = 'HTSeq–Counts'". Read counts were normalized using the "rpkm" method from the "edgeR" package ("log = FALSE, prior.count = 1") based on maximum transcript length per gene, obtained via biomaRt and the Ensembl database. Segmented copy number profiles for each sample were downloaded from the GDC data portal. Gene copy number status was defined as the maximal absolute $\log_2$ ratio among segments spanning the gene in a given sample. Genes with both copy number and expression values assigned were retained.

To focus on genomic regions subject to copy number changes recurrent enough to indicate selection, TCGA GISTIC results[4] were used (obtained from gdac.broadinstitute.org, accessed 4 July 2017) and broad copy number changes with $q$ values < 0.05 were kept. To focus on genes altered at relevant frequencies and more likely to be part of any minimal region of overlap, genes with absolute $\log_2$ copy number ratio < 0.2 were filtered out and only genes with an alteration

frequency in the upper third quartile per chromosome arm event were retained. The third quartile was chosen, rather than a stricter threshold, as some regions may be subject to low-frequency focal events of a random or artifactual nature.

Genes altered in expression in tandem with the copy number changes were determined using linear regression between copy number and expression, adjusting for tumor purity according to published estimates[68]. Genes with too low expression variance to test (where regression failed to converge) were removed. Univariate survival tests with Cox regression ("coxph" from the "survival" R package[69]) were carried out against clinical data downloaded using TCGAbiolinks, using the variables "vital_status", "days_to_death", and "days_to_last_follow_up".

For the metastasis data set, segmented copy number values were mapped to gene names as described above and converted to $\log_2$ ratios. Values of zero prior to transformation were set to the lowest observed non-zero copy number value. Gene expression values were normalized with RPKM as described and batch-corrected ("removeBatchEffect" from the "limma" R package[70]). Genes with both copy number and expression values assigned and also present in the filtered TCGA data set were retained. Associations between expression and copy number were assessed as for the TCGA data set, considering sample purity. $p$ values from associations in the TCGA data and metastasis data set were combined using Fisher's method and FDR adjusted using the Benjamini–Hochberg method. Candidates with $q$ values < 0.05, independent raw $p$ values < 0.05 in each data set and correlations consistent with the direction of the assessed copy number event were retained. Candidates in regions with more samples harboring gains than losses were retained as candidates of gains and vice versa.

To assess the extent to which a given gene may have a wider impact on cellular behavior, protein–protein interactions with experimental evidence defined in the HPRD database (accessed using the iRefR R package)[71,72] were used. Candidates were then ranked by the number of HPRD connections, followed by whether any univariate survival associations existed ($p < 0.05$) that implied worse survival consistent with the nature (gain or loss) of the copy number event assessed. This way, survival associations were placed a low weight, with the motivation that such associations are easily confounded by multiple genomic and clinical factors.

**siRNA screen.** In vitro knockdown of selected genes was performed using siRNA in UM22, MP-41, and 92-1 cells. Transient transfection was performed with mock siRNA (control), a positive control siRNA (*GNAQ*) or a pool of four siRNA per gene of interest. The siRNA duplexes were purchased from Dharmacon (Thermo Fisher Scientific, Waltham, MA, USA) and the lipid based transfection was performed with Lipofectamine-RNAiMAX (Thermo Fisher Scientific, Waltham, MA, USA) using 1 pmol of siRNA per well of a 96-well plate as per the guidelines provided by manufacturer. The RNA-Lipid complex was made in Opti-MEM Reduced Serum medium. The cells were seeded in triplicates in black 96-well plates (Corning). Post transfection (72 h, 96 h, and 96 h, respectively), cells and viability were monitored with ATP measurement using CellTiter-Glo Assay (Promega). Luminescence was measured with GloMax Discover plate reader (Promega). In parallel, manual cell counting was performed using Trypan blue staining of cells obtained from transfections in 12-well format.

**Generation of PDX models and a BAP1-deficient UM cell line.** Animal experiments were performed in accordance with E.U. directive 2010/63 (Regional animal ethics committee of Gothenburg approvals #36-2014 and #1183-2018). Cryopreserved biopises were thawed and single cells were transplanted into the flank (UM22) or the liver (UM9) of immunocompromised, 6–8-week-old female non-obese severe combined immune deficient interleukin-2 chain receptor γ knockout mice (NOG mice; Taconic, Denmark) to form xenografts. The mice were housed in the pathogen-free animal facility of University of Gothenburg. Mice were kept in cages with individual ventilation at ambient temperature (21–23 degrees Celsius) and 20–40% humidity. Mice were given free access to food and water. The dark–light cycle was 12 h dark and light, respectively (dark 7 pm to 7 am). Tumors were analyzed by immunohistochemistry using clinically used antibodies against Melan-A (clone A103, catalog: IS63330-2, ready-to-use solution), PMEL (clone: HMB45, catalog: GA05261-2, ready-to-use solution for Autostainers) and S100 (clone: IR504, catalog: IS50430-2, ready-to-use solution), purchased from Agilent/Dako. For generation of a cell line, a PDX tumor was minced and seeded at high density into a 5 cm culture plate in RPMI medium supplemented with 10% fetal bovine serum. Surviving cells were expanded and transduced with either a retrovirus expressing HA-tagged BAP1 or a control retrovirus (MSCV-IRES-GFP), both of which were made using plasmids from Addgene. $n = 3$ independently grown replicates from each were then characterized by RNA-seq. The UM22 cell line generated is available upon request. BAP1 protein expression was measured with IHC using an antibody purchased from Santa Cruz Biotechnology (clone: C-4; catalog: sc-28383) with dilution 1:1000 and with western blot using dilution 1:100. Beta-actin antibody used for Western blot was purchased from Sigma (clone AC-15, catalog: A1978, dilution 1:10000).

**Differential gene expression analysis.** RNA-seq data were aligned and quantified as described. Differential expression was assessed using DESeq2, with the parameter "alpha = 0.05". Gene set enrichment analysis was carried out with the R package "fgsea"[73], with gene sets obtained from MSigDB[42], using the parameters

"minSize = 0, maxSize = 10000, nperm = $10^7$". Genes and gene sets with $q < 0.05$ were considered statistically significant.

**RT-qPCR validation**. RNA was extracted from the indicated cell lines with Nucleospin RNA II kit (Macherey-Nagel), and converted to cDNA using iScript cDNA synthesis kit (Bio-Rad). qPCR was performed using 2× qPCR SyGreen Mix (PCR Biosystems) and the CFX Connect Real-Time System (Bio-Rad). Data analysis was performed by comparing ΔΔCT values using Ubiquitin as a reference gene. $n = 2$ technical replicates were used.

**Protein extraction for proteomic analysis**. Cell pellets were lysed by sonication in 400 μl in 2% sodium dodecyl sulfate and 50 mM triethylammonium bicarbonate (TEAB). Samples were centrifuged at 13,000 rpm for 10 min and the supernatants were transferred to clean tubes. Protein concentration of lysates was determined using Pierce BCA Protein Assay Kit (Thermo Scientific) and the Benchmark Plus microplate reader (BIO-RAD) with bovine serum albumin solutions as standards.

**Tryptic digestion and tandem mass tag (TMT) labeling**. Aliquots containing 30 μg of total protein were taken from each sample and reduced at 56 °C for 30 min in the lysis buffer with DL-dithiothreitol at 100 mM final concentration and incubated. The reduced samples were processed using the modified filter-aided sample preparation method[74]. In short, reduced samples were diluted to 400 μl by addition of 8 M urea, transferred onto Nanosep 30k Omega filters (Pall Life Sciences) and washed two times with 200 μl of 8 M urea. Alkylation of the cysteine residues was performed using 10 mM methyl methanethiosulfonate diluted in digestion buffer (1% sodium deoxycholate (SDC), 50 mM TEAB) for 20 min at room temperature and the filters were then repeatedly washed with digestion buffer. Trypsin (Pierce Trypsin Protease, MS Grade, Thermo Fisher Scientific) in digestion buffer was added in a ratio of 1:100 relative to total protein mass and the samples were incubated at 37 °C for 3 h; another portion of trypsin (1:100) was added and incubated overnight. The peptides were collected by centrifugation and labeled using TMT reagents (Thermo Scientific) according to the manufacturer's instructions. The labeled samples were combined; pooled samples were concentrated using vacuum centrifugation, and SDC was removed by acidification with 10% trifluoroacetic acid and centrifugation. The combined TMT-labeled sample was fractionated into 40 primary fractions by basic reversed-phase chromatography using a Dionex Ultimate 3000 UPLC system (Thermo Fischer Scientific). Peptide separations were performed using a reversed-phase XBridge BEH C18 column (3.5 μm, 3.0 × 150 mm, Waters Corporation) and a linear gradient from 3% to 40% solvent B over 17 min followed by an increase to 100% B over 5 min. Solvent A was 10 mM ammonium formate buffer at pH 10.00 and solvent B was 90% acetonitrile, 10% 10 mM ammonium formate at pH 10. The primary fractions were concatenated into final 20 fractions (1 + 21, 2 + 22, … 20 + 40), evaporated and reconstituted in 15 μl of 3% acetonitrile, 0.2% formic acid for nLC MS analysis.

**Liquid chromatography–mass spectrometry/mass spectrometry analysis**. The fractions were analyzed on an Orbitrap Fusion Lumos Tribrid mass spectrometer, equipped with a FAIMS Pro Source and interfaced with Easy-nLC1200 liquid chromatography system (both Thermo Fisher Scientific). Peptides were trapped on an Acclaim Pepmap 100 C18 trap column (100 μm × 2 cm, particle size 5 μm, Thermo Fisher Scientific) and separated on an in-house packed analytical column (75 μm × 30 cm, particle size 3 μm, Reprosil-Pur C18, Dr. Maisch) using a linear gradient from 5% to 33% B over 77 min followed by an increase to 100% B for 3 min, and 100% B for 10 min at a flow of 300 nL min$^{-1}$. Solvent A was 0.2% formic acid in water and solvent B was 80% acetonitrile, 0.2% formic acid. Two experiments were performed in parallel during the 90 min separation for all MS scans and sequential MS2 and MS3 scans. One experiment using a FAIMS Compensation voltage (CV) of −30 V and a second one using a CV of − 50 V. MS scans was performed at 120,000 resolution, m z$^{-1}$ range 375–1375, MS/MS analysis was performed in a data-dependent, with top speed cycle of 3 s for the most intense doubly or multiply charged precursor ions. Most intense precursors were fragmented in MS2 by collision induced dissociation at 35 collision energy with a maximum injection time of 50 ms, and detected in the ion trap followed by multinotch (simultaneous) isolation of the top 10 MS2 fragment ions, with m z$^{-1}$ 400–1400, selected for fragmentation (MS3) by higher-energy collision dissociation (HCD) at 65% and detection in the Orbitrap at 50,000 resolution, m z$^{-1}$ range 100–500. Precursors were isolated in the quadrupole with a 0.7 m z$^{-1}$ isolation window and dynamic exclusion within 10 ppm during 45 s was used for m z$^{-1}$ values already selected for fragmentation.

**Proteomic data preprocessing**. Identification and relative quantification was performed using Proteome Discoverer v. 2.2 (Thermo Fisher Scientific). The reference *Homo sapiens* database was downloaded from SwissProt (September 2019). The database search was performed using the Mascot search engine v. 2.5.1 (Matrix Science, London, UK) with MS peptide tolerance of 5 ppm and fragment ion tolerance of 0.6 Da. Tryptic peptides were accepted with 0 missed cleavages only; methionine oxidation was set as a variable modification, cysteine methylthiolation, TMT-6 on lysine and peptide N-termini were set as fixed modifications. Percolator was used for peptide-spectrum match validation with the strict FDR

threshold of 1%. Quantification was performed in Proteome Discoverer 2.2. TMT reporter ions were identified in the MS3 HCD spectra with 3 mmu mass tolerance, and the TMT reporter intensity values for each sample were normalized within Proteome Discoverer 2.2 on the total peptide amount. Only the unique identified peptides were taken into account for the relative quantification.

**Differential protein expression analysis**. Normalized abundances of identified proteins, on log$_2$ scale, were compared between *BAP1*-reintroduced cells and empty vector-treated control cells with the R package limma (v. 3.40.6) and $p$ values were adjusted using the Benjamini–Hochberg method. $n = 3$ independently grown samples of cells derived from either the case or control cell lines were used, respectively. Proteins with $q < 0.05$ (Benjamini–Hochberg correction) were considered statistically significant. Gene set enrichment analysis was carried out as for the corresponding RNA-seq comparison.

**Comparisons between *BAP1* mutant and wild-type TCGA UM tumors**. TCGA UM gene expression data were downloaded and normalized as described above. A list of *BAP1* mutated tumors was compiled from earlier publications[4,46] and three additional likely cases (Supplementary Fig. 5i) were identified from exome data downloaded from GDC (accessed 11 September 2019). $n = 40$ biologically independent samples were included in each condition. Differential expression was assessed with Wilcoxon rank-sum tests and genes with $q < 0.05$ after Benjamini–Hochberg correction were considered statistically significant.

**Single-cell RNA-seq analysis of immune infiltrates**. Small pieces of tumor biopsies were cultured for 2 weeks in RPMI medium containing 10% human serum and 6000 U ml$^{-1}$ IL-2. γTIL cultures were then cryopreserved before use. Two days before performing the single-cell experiments, γTIL cultures were thawed. Cells were counted and 7000 cells were injected into a single-cell library preparation instrument (10x Genomics). The steps following were performed using the Single Cell V(D)J kit according to the kit description (10x Genomics). V(D)J libraries were sequenced on an Illumina MiSeq, whereas the gene expression libraries were run on an Illumina NextSeq 500. Single-cell transcriptomics data were aligned against the hg38 reference genome and preprocessed using the Cellranger pipeline (v. 2.1.1) provided by 10x Genomics. Expression levels were estimated using the Cellranger "count" function, with default parameters. TCR chain assembly was also performed using the Cellranger pipeline, using default parameters. t-SNE maps for each sample were generated using Seurat (v. 3.1.3) and cell types were inferred using the approach by Zheng et al. (https://github.com/10XGenomics/single-cell-3prime-paper)[75], with some modifications: correction of a code error that misclassified some CD4$^+$ cells; reclassification of cells classified as memory CD4$^+$ cells not expressing *CD4* but *CD8A* as the closest matching non-CD4$^+$ cell type; predicted non-T cells that express TCRs as closest matching T-cell type; predicted dendritic cells that express *CD3G* or *NCAM1* as the closest non-dendritic cell type. Doublet cells were defined as those predicted as such by DoubletFinder[76] (v. 2.0.2, default parameters) and additionally as those expressing more than two alpha or beta chains. TCR clonotype diversity (normalized entropy) was assessed for CD8$^+$ cells using clonotype frequency and the "diversity" function (type = "e") from the "diverse" R package[77], giving the Shannon entropy, which was then normalized by dividing by the logarithm of the number of unique clonotypes for each sample[78].

**Flow cytometry**. Single-cell suspensions from cryopreserved tumor biopsies and γTILs were surface stained for 30 min in RT. The following antibodies were used for surface staining: CD3 (clone: HIT3a, catalog: 300306, lot: B274310, dilution: 1:200, fluorochrome: AF488), CD4 (clone: A161A1, catalog: 357414, lot: B238830, dilution: 1:200, fluorochrome: PerCP/5.5), CD8 (clone: HIT8a, catalog: 300920, lot: B256905, dilution: 1:200, fluorochrome: AlexaFluor700), CD45 (clone: 2D1, catalog: 368516, lot: B251494, dilution: 1:200, fluorochrome: APC-Cy7), CD69 (clone: FN50, catalog: 310933, lot: B251799, dilution: 1:200, fluorochrome: BV650), CTLA-4 (clone: BNI13, catalog: 369603, lot: B242857, dilution: 1:100, fluorochrome: PE), PD-1 (clone: EH12.H7, catalog: 329920, lot: B255122, dilution: 1:200, fluorochrome: BV421), TIGIT (clone: A15153G, catalog: 747839, lot: 8267685, dilution: 1:100, fluorochrome: BV711) and TIM-3 (clone: F38-2E2, catalog: 345031, lot: B241708, dilution: 1:100, fluorochrome: BV785) from BioLegend and CD39 (clone: eBioA1, catalog: 25-0399-41, lot: 4329374, dilution: 1:200, fluorochrome: PE-Cy7) from eBioscience. For detection melanoma antigen specific CD8 T cells, cells were surface stained for 45 min in 37 °C using Melanoma Dextramer Collection 1 kit from Immudex (catalog: RX01, lot: 20190611-KB). Dead cells were excluded from the analysis using Live/Dead Aqua (Invitrogen). Flow cytometry data were acquired using BD Accuri C6 (BD Biosciences), BD LSRFortessa X-20 (BD Biosciences) or BD FACSARIA FUSION (BD Biosciences) and analyzed using FlowJo software (FlowJo LLC). The gating strategy is shown in Supplementary Fig. 6a, b. TILs tested for dextramer binding were those from HLA-A*02-positive tumors (Supplementary Data 7), for which commercial dextramers with MART-1, gp100 and NY-ESO-1 were available: UM1, UM2, UM9, UM11, UM13, UM21, UM22, UM24, UM25, UM26, UM27, UM29, UM30, UM31, and UM32. Samples with TILs that were reactive towards these antigens are shown in Fig. 4b, whereas the remaining ones did not show signs of reactivity.

**Ethics**. The patients received oral and written information and signed the informed consent agreement according to the ethical approval at the Göteborg Human ethical review board (#289-12 and #144-13 and #44-18). Animal handling: regional animal ethics committee of Gothenburg approvals #36-2014 and #1183-2018.

**Reporting summary**. Further information on research design is available in the Nature Research Reporting Summary linked to this article.

## Data availability

Whole-genome, exome, and RNA sequence data generated for this study have been deposited at the European Genome-Phenome Archive, which is hosted by the EBI and the CRG under accession numbers EGAS00001004296 and EGAS00001003026. Single-cell transcriptomics and TCR data are available at ArrayExpress with the identifier E-MTAB-8846. TCGA data used in this study are available from the GDC Data Portal (https://portal.gdc.cancer.gov/) under restrictions of controlled access for raw data. Mass spectrometry proteomics data generated for this study have been deposited to the ProteomeXchange Consortium via the PRIDE partner repository with the data set identifier PXD017743. Figures with associated raw data are Figs. 1–4, Supplementary Figs. 1–3, and Supplementary Figs. 5–8. Mutation data from the COSMIC database can accessed at https://cancer.sanger.ac.uk/cosmic/download and mutational signature data at http://cancer.sanger.ac.uk/cancergenome/assets/signatures_probabilities.txt. Data from the Human protein reference database (HPRD) was accessed through the iRefR R package and the database can be downloaded at http://hprd.org/download. Reference protein data from SwissProt was accessed through the Mascot software and the database can be downloaded from https://www.uniprot.org/downloads. Databases formatted for use with ANNVOAR, including ESP6500, 1000 Genomes and dbSNP, can be downloaded by following the instructions at http://annovar.openbioinformatics.org/en/latest/user-guide/download/. MSigDB gene sets can be accessed at https://www.gsea-msigdb.org/gsea/msigdb/genesets.jsp. Data from the Genome Aggregation database (gnomAD) can be accessed at ftp://ftp.broadinstitute.org/bundle/Mutect2/af-only-gnomad.raw.sites.b37.vcf.gz.

## Code availability

Code is available at https://bitbucket.org/jowkar/um.

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

## Acknowledgements

The results published here are in part based upon data generated by the TCGA Research Network (cancergenome.nih.gov/). We thank Sofia Stenqvist for animal care, Ola Nilsson and Gülay Altiparmak for histology, Carina Karlsson for technical assistance and Therese Bengtsson, Valerio Belgrano and surgeons in the SCANDIUM trial for patient sampling and registrations. Quantitative proteomic analysis was performed by Britt-Marie Olssen and Evelin Berger at the Proteomics Core Facility of Sahlgrenska Academy, University of Gothenburg. The GeneCoreSU and SciLifeLab facilities are acknowledged for sequencing services, partly financed by a National Genomics Initiative grant to J.A.N. Other funding sources included Knut and Alice Wallenberg Foundation, Cancerfonden, Vetenskaps-rådet, Sjöbergstiftelsen, Familjen Erling Perssons stiftelse, Wilhelm & Martina Lundgrens Vetenskapsfond, the Assar Gabrielsson foundation and Västra Götalandsregionen.

## Author contributions

Conceptualization: J.K., L.M.N., R.O.B., J.N. Data curation: J.K., H.J., R.O.B. Formal analysis: J.K. Funding acquisition: J.K., E.L., P.L., R.O.B., J.N. Investigation: J.K., L.M.N., S.M., S.A., G.S., V.S., E.F., B.E. Methodology: J.K., L.M.N., J.N. Project administration: JN. Resources: U.S., C.A., L.N., P.L., E.L., R.O.B., J.N. Supervision: J.N. Validation: L.M.N., G.S., V.S., E.F. Visualization: J.K., L.M.N., J.N. Writing—original draft: J.K., JN. Writing—review & editing: J.K., L.M.N., S.M., L.N., P.L., E.L., R.O.B., J.N.

## Competing interests

The authors declare no competing interests.

## Additional information

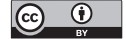

