## [Peer Review File · Nature Communications]

Reviewers' Comments:

Reviewer #1:

Remarks to the Author:

Karlsson et al. analyze biopsy samples from metastatic uveal melanoma patients for genomic alterations. Treatment of metastatic uveal melanoma is an unmet need in the melanoma field and there are few studies that focus on metastatic tumor analysis. The analysis of metastatic samples in this disease is significant and important. The study adds to the recent Shain et al., manuscript that studied patient-matched primary and metastatic samples (Nature Genetics). Other studies have analyzed effects of BET inhibitors on the expression of growth factor receptors in biopsy samples from metastatic uveal melanoma patients (Chua et al., EMBO Mol Med). While, overall, this manuscript is rather descriptive it does contain important new information especially regarding the characterization of TIL from metastatic lesions.

My main comments are that the functional effects of 8q genes needs clarification since this is the most mechanistic section of the manuscript, whether BAP1 targets are regulated at the protein levels should be determined, and the association between up-regulation of immune checkpoints in the tumor cells and the presence of distinct T cell populations could be further explored given the novelty of these studies.

Other comments:

1. Please clarify where the PRAME expression data is shown.
2. A number of mutation are identified at low frequency including YEATS, ZMAT3, TET1 and TET2. Are these found in primary uveal melanoma?
3. Fig. 2h needs statistical analysis. It is unclear whether this cell line is dependent on GNAQ. Also, the level of knockdowns need to be shown. The lack of effect with some of the siRNA is non-interpretable in the absence of such data.
4. In experiments associated with Figure 3, the levels of BAP1 re-expression relative to wild-type BAP1 expressing cells needs to be shown.
5. Fig 3e- qRT-PCR experiments do not seems to confirm regulation of RAB31.
6. Are TIM-3 and TIGIT ligand levels higher in BAP1 mutant primary tumors versus WT BAP1 tumors from the TCGA dataset and are these targets regulated at the protein level in a BAP1-dependent manner within the re-expressing cell lines?

Reviewer #2:

Remarks to the Author:

The authors characterize 32 cases of metastatic uveal melanoma (UM) by performing whole genome sequencing as well as immune cell profiling. This study thus represents a rare insight into UM in the metastatic setting. Unfortunately, the study remains largely descriptive and limits a mechanistic exploration to a single cell line. From an immunologists point of view, the analysis of the TIL populations carries little weight, as it is performed after in vitro expansion and does therefore not reflect the true characteristics of T cells within the tumor.

Major comments:

- The authors evaluate the functional impact of BAP1 loss on tumor phenotype by reintroducing the gene in a xenograft-derived cell line. While the findings are interesting, independent confirmation in at least one additional cell line would be desirable. It is also not clear how well the cell lines' transcriptome reflects that of the original tumor. The authors need to demonstrate that the cell line used is a meaningful representation of the in vivo situation.
- Tumor-infiltrating lymphocytes (TILs) are analyzed by flow cytometry (ex vivo and after expansion), as well as by TCR and RNA sequencing after expansion. The fact that TILs underwent a two week expansion in the presence of high-dose IL-2 precludes any meaningful interpretation of the single-cell sequencing analysis. While the authors claim that CD4/CD8 proportions are well-

maintained, this is not true for at least half of the patients. Figure S9 further indicates that phenotypic changes will occur during culture.

Minor comments:

- Analysis of T cell infiltrates in patients (cutaneous UM) that have recently undergone chemotherapy should be interpreted with caution, as lymphocytes are likely diminished as a result of cytotoxic therapy. The authors should comment on how/if this issue was addressed.
- Dextramer staining is performed for a subset of patients. The kit used includes only HLA-A2 multimers, but no information on the selection and HLA-types of the analyzed patients is given. Data supporting HLA-A2 expression in the selected patients should be included.
- The materials and methods section describes the definition of doublets from single cell data, however it is not clear how doublet events are handled. Optimally, these should be excluded from down-stream analysis, though in Figure 4d they appear to be listed alongside singlet events and analyzed as separate groups, though the meaning or significance of this is not made clear. In addition, please note that about 30% of T cells will express two alpha chains – this characteristic can thus not be used for doublet exclusion.
- The plots depicted in Figure 4d suggest that large clonotypes significantly contribute to the clustering of the single cell data. This bias should be removed to better enable interpretation of the TIL phenotypes.
- The description of the correlation of clonotypes with cytotoxicity and exhaustion markers is difficult to follow, both in the materials and methods section and in the figure legend supporting Figure S8. The figure legend further states that the phenotype is derived from flow cytometry, rather than transcriptomics. Is this correct? If so, i) what is the exact phenotype used and ii) is the result supported by transcriptomics data?
- Figure 4: the legend must reveal that analysis expanded TILs is shown
- Figure 4: To allow better comparison between samples, Shannon's entropy is often normalized to the observed diversity.
- Figure S7: no legend exists for panel c) and d)
- Figure S9: the number of (acquired) CD4 and CD8 cells within a single patient appears to be different between the plots showing PD1/TIGIT and FSC/CTLA4. Please comment.
- Figure S10: no such figure was uploaded

Reviewer #3:

Remarks to the Author:

The paper "Molecular profiling of driver events and tumor-infiltrating lymphocytes in metastatic uveal melanoma" presents data on 32 metastases from uveal melanoma. Eventhough the authors uses GWAS technique the basically confirms what is already known regarding mutations and CNV 1,2. The data on transfection with functional BAP1 is really novel and very interesting, but is only performed in one sample. Similarly is the experiment on siRNA blockage of proliferation responsible genes very interesting, but again only in one selected sample. In 8 samples are tumor infiltrating lymphocytes analyzed, but no clear conclusion from these experiments is given.

Setup: The authors compare metastases from liver and skin and from both anterior and posterior part of the eye. It can be questioned whether all parts of uvea are comparable. AJCC distinguishes between iris melanomas from the anterior part of the eye and choroidal/ciliary body melanomas from the posterior part of the eye. The authors do find discrepancies that confirms this clinically separation. From figure 1b it can be argued that metastases from the skin is not comparable to metastases from the liver. 4/6 metastses from the skin have no deletions in chr 3 compared to 0/21 metastases from posterior uveal melanoma and 1/1 metastasis from anterior uveal melanoma, indicating that posterior uveal melanoma has a "non-skin-like" type in contrast to skin and iris.

Why was all genetic experiments not performed on all samples? Could there be a bias introduced affecting the results?

The preparation of cm3 + cm 4 and ocular specimens are not described in the paper

Abstract:

The authors states “..Lack of clinical rationale to biopsy metastatic disease”, which is clinically not correct as 9-18% of patients have metastatic disease from second primary cancer.

The authors only investigate 2 matched primary tumors with their metastases so the statement “ .. some of which were not altered in matched primary eye tumors.” gives the impression that this match was done for more tumors. From figure 2C, which basically is the only information on this statement, there is a fairly good correlation between primary tumor and their metastasis and all clinically relevant changes can be found in both samples.

Introduction:

The citation for PRAME does only state on the use for prognostication for Class 1 tumors, as class 2 tumors always have poor prognosis. So the binary separation is only for class 1 tumors.

Results:

P4: Did the authors look for functional BAP1 with IHC3?

P4: UM28 had polyploidy, could the homozygotic loss of BAP1 be due to isodisomy?

P4: Again the use citation only states om PRAME and class 1 tumors.

P6: Comparison between TCGA material and the metastases from this material seems not appropriate as the TCGA contained data from both metastatic and non-metastatic tumors. TCGA showed a significant difference between the metastatic and non-metastatic groups. Therefore all findings would automatically be significant, due to the basic difference (Figure 2b)

P6: The mentioned survival, is that given from the HPRD database or is it the observed survival?

P6: the siRNA are really interesting, but only one, highly different from the other samples was used, and therefore conclusions should be made very cautiously.

P7: were the transfected cells stained for IHC BAP1 to show functional BAP1?

Discussion:

P9: biopsy of metastasis is meaningfull – see above

P9: Comparison of TCGA data with this dataset – see above

Figure 1a: would improved if number of samples were given for each sub-experiment

Figure 1e: seems some information is missing (cm3, cm4, um25, um28, um29, um30 and um31).

Why was um4, um5, um6, um15, um16, um22, um23, um25 and um31 not analyzed?

Figure 2a: um19 is stated in figure 1b to have large deletion in chr.3 but in this figure it seem chr.3 is normal. Is the low purity (figure S2) causing misinterpretation of this sample?

Figure 2f: What does the inner and outer circle represent?

Figure S3: the labelling of panels is not unambiguous.

Figure S5: Is the panels showing IHC stains from 2 PDX experiments or 2 different area in the PDX or the patient metastasis and PDX?

Figure S9: c+d panel no explained in test

Figure S10: c+d panel is nonexisting

1. Royer-Bertrand B, Torsello M, Rimoldi D, et al. Comprehensive Genetic Landscape of Uveal Melanoma by Whole-Genome Sequencing. *Am J Hum Genet* 2016; 99(5): 1190-8.
2. Shain AH, Bagger MM, Yu R, et al. The genetic evolution of metastatic uveal melanoma. *Nat Genet* 2019; 51(7): 1123-30.
3. Patrone S, Maric I, Rutigliani M, et al. Prognostic value of chromosomal imbalances, gene mutations, and BAP1 expression in uveal melanoma. *Genes Chromosomes Cancer* 2018; 57(8): 387-400.

Responses

Please allow us to express our gratitude for the many excellent suggestions for improvements of our manuscript. Our responses and our alterations are described below:

Reviewer 1

1. *The functional effects of 8q genes needs clarification since this is the most mechanistic section of the manuscript*

Two additional cell lines have been investigated with respect to knockdown of these genes (revised **Fig. 2h, S3c**). In addition, siRNA knockdown efficiency has been assessed, the results of which are shown in supplementary **Fig. S3d**. Statistical analysis has also been done for the knockdown proliferation/viability experiments, and q -values for the effect on each targeted gene have been added to **Fig. 2h**. Note also that the genes we investigated are chosen among the highest-ranked genes for each of the chromosomal regions of gain, and are not limited to those on 8q. We are investigating further consequences on downstream cellular signaling upon perturbing these genes, but this is a complex aspect to provide conclusive details on here that we will have to leave to a follow-up study. We do note, however, that several of the genes mentioned are related to *PTK2* (FAK)-associated signaling pathways (notably *RAC1* and *MAPK14*) which have been proposed to regulate *YAP1* to drive proliferation (which also appears to be downstream of *GNAQ* and *GNA11*)¹.

2. *Whether BAP1 targets are regulated at the protein levels should be determined*

We have performed mass spectrometry-based proteomics on the *BAP1*-null and *BAP1*-reintroduced cell lines used for this experiment. We found that 1943 genes/proteins were significantly differentially expressed in the same direction in both the RNA and proteomics data (new **Fig. S5e**). We additionally confirmed that the larger Class II low/high-related gene sets from MSigDB were enriched in a similar way as with the RNA-seq data (**Fig. S5f**) and that three of the four ligand-related genes were significantly altered in a consistent manner also at the protein level (**Fig. S5g**). See also the response to reviewer 1, comment 9.

3. *The association between up-regulation of immune checkpoints in the tumor cells and the presence of distinct T cell populations could be further explored given the novelty of these studies.*

We would like to explore this further, but the single-cell RNA-seq data is limited in terms of sample size (eight samples) and also in that all those samples are *BAP1* mutated, so we would not be able to compare differences in TIL communities between *BAP1*-mutated and wild-type tumors and the samples are also too few and diverse in terms of tumor composition to meaningfully assess correlations to bulk RNA-seq data. We (and others²) have compared the expression of immune checkpoint receptors between TCGA samples with and without *BAP1* mutation and found there to be significantly greater expression of these as well as the ligands we discuss in *BAP1* mutated tumors (Figure for Reviewers **Fig. R1** at the end of this rebuttal and **Fig. S5k**). Greater expression of the receptors in TCGA *BAP1*-mutated bulk tumor material can however also be explained by greater T cell infiltrates overall (CD3G, CD8A and CD4 expression levels shown in **Fig. R1**), although this would not explain upregulation of the ligands, which could be contributed at least in part via expression changes that occur following *BAP1* mutation. Given that the differences exist in TCGA primary tumors, this seems to indicate that these changes are not caused by different conditions particular to a remote metastatic site (the liver has a unique immune biology, for instance).

4. *Please clarify where the PRAME expression data is shown.*

PRAME expression has been added to **Fig. S1e** and the text has been updated to reference this panel.

5. *A number of mutation are identified at low frequency including YEATS, ZMAT3, TET1 and TET2. Are these found in primary uveal melanoma?*

We have matching primary material sequenced for UM11, which had the *TET2* mutation, and do not detect it in the primary, although read count was low (12 reads) at that position in the primary material, so we cannot exclude that it could be present. Comparisons between matched primary-metastasis pairs we have sequenced have also been added to **Table S2**, although it should be cautioned that the eight primary tumors are from FFPE material and have poor coverage in addition to varying levels of purity, which contributes to lack of detection of a number of possibly shared mutations (this caveat is also explained in the same table). For the other three genes we do not have matching primary tumors available to compare, but we have instead also compared the frequencies of mutations in all genes (these four as well as all others) between our cohort and TCGA UM tumors, the latter of which are all primary. The comparison is shown in **Fig. S1f-g** (which includes genes that had at least 1 mutation in TCGA UM). Neither *YEAST2*, *ZMAT3*, *TET1* nor *TET2* were mutated in any samples in TCGA primary UMs, which may support that they could be late events.

6. *Fig. 2h needs statistical analysis. It is unclear whether this cell line is dependent on GNAQ. Also, the level of knockdowns need to be shown. The lack of effect with some of the siRNA is non-interpretable in the absence of such data.*

Statistical analysis based on two-way permutation-based ANOVA (to take into account both cell line and targeted gene) has been performed for these experiments, and *q*-values for the effect on each targeted gene independent of cell line have been added **Fig. 2h** and **Fig. S3c** (the set of genes were split to show primary candidates from **Fig. 2f** in **Fig. 2h** and secondary genes in **Fig. S3c**, but FDR-adjusted *p*-values were calculated on all genes together). siRNA knockdown efficiency estimates have also been determined and added in **Fig. S3d**. UM22 has a *GNAQ* Q209L hotspot mutation that is known to be oncogenic³ and we expect detrimental effects on cell proliferation upon knockdown of this gene, motivating it as a control. This also seems to be the case, according to the results of the experiment and since knockdown levels for *GNAQ* appeared to be ~70-75% (**Fig. S3d**). One of the other two cell lines used for comparison, 92-1, also has a *GNAQ* Q209L mutation, whereas MP41 does not, but rather a *GNA11* mutation⁴.

7. *In experiments associated with Figure 3, the levels of BAP1 re-expression relative to wild-type BAP1 expressing cells needs to be shown.*

We have performed IHC comparing BAP1 levels in the two conditions and added in this to **Fig. 3b-c**. Expression levels based on RNA-seq are also now shown in **Fig. S5c-d**.

8. *Fig 3e- qRT-PCR experiments do not seems to confirm regulation of RAB31.*

The RNA-seq analysis found 9/12 of these genes to be significant, with *RAB31*, *ID2* and *EIF1B* being the three that were not being significant. The qRT-PCR analysis was done for all 12 genes regardless and confirms lack of regulation for *RAB31*. This has now been clarified at in the main text.

9. *Are TIM-3 and TIGIT ligand levels higher in BAP1 mutant primary tumors versus WT BAP1 tumors from the TCGA dataset and are these targets regulated at the protein level in a BAP1-dependent manner within the re-expressing cell lines?*

We have now compared *BAP1*-mutated and *BAP1* WT tumors in TCGA UM and found that three of the four ligand-related genes in **Fig. 2h** were significantly elevated in *BAP1* mutated tumors, the exception being *PVRL2* (**Fig. S5k**). We have also performed a mass spectrometry proteomics analysis to compare the *BAP1*-mutated and *BAP1*-reintroduced cell lines and also here found that three out of them differed in the same way as on RNA-seq, with the exception again being *PVRL2* (**Fig. S5g**).

Reviewer 2

1. *Unfortunately, the study remains largely descriptive and limits a mechanistic exploration to a single cell line.*

Our focus has mainly been on describing genomic alterations that contribute to define the malignant state of this cancer and we have followed up mutation of the tumor suppressor *BAP1* experimentally, which has a less well-defined role in tumorigenesis and metastasis than the otherwise frequently mutated *GNAQ* and *GNA11*. In addition, we also described genes that are altered as a consequence of

recurrent copy number changes and performed knockdown experiments on 17 of them to study effects on cell proliferation and viability.

We have expanded on the latter experiment by also testing the same genes on two additional UM cell lines: 92-1 and MP41. Those results are shown in **Fig. 2h** and **Fig. S3c-d** and have also been discussed in response to reviewer 1, comments 1 and 6.

For the *BAP1* reintroduction experiment, we used UM22, a cell line we derived ourselves from one of the tumors profiled in this study. During the six years we have worked with building a biobank of metastatic uveal melanoma biopsies UM22 is also the only *BAP1*-mutated UM cell line that we have been able to grow. Three other cell lines we grow routinely in the lab (92-1, MEL202 and MP41) all express functional *BAP1*. There are a few *BAP1* deficient cell lines described with doubling times of over 100 hrs but none are like ours which grow just as fast as the *BAP1* proficient cell lines. This in itself is interesting and we would be very interested in finding out why preferentially tumors lacking *BAP1* mutations are able to establish themselves as stable cell lines. We have some indications that this could be related to where the mutation occurs in the *BAP1* sequence. UM22 has a frame-shift deletion that occurs close to the end of the protein, such that it eliminates the nuclear localization signal domain, but retains the other parts of the protein. This is the case for only two other tumors in our cohort and it could be that remaining functionality of the protein performed in the cytoplasm could have a role⁵. IHC performed on the cell line (**Fig. 3b-c**, **Fig. S4b-c**) shows loss of nuclear staining, but remaining cytoplasmic staining. Nuclear staining is restored in the *BAP1*-reintroduced version of the cell line (**Fig. 3b-c**). We are experimenting with CRISPR-induced mutations in different parts of the gene to compare differences in effect.

Regardless, Harbour et al.^{6,7} have previously performed the inverse variant of the experiment, with knockdown of *BAP1* in *BAP1* wild-type 92-1 cells and findings in our cell line are compatible with the changes in the differentiation and proliferation markers they describe as altered in the 92-1 cell line, supporting our overall conclusion that *BAP1* mutation is responsible for a subtype-switch.

2. *From an immunologists point of view, the analysis of the TIL populations carries little weight, as it is performed after in vitro expansion and does therefore not reflect the true characteristics of T cells within the tumor.*

This study is largely based on biopsies from surgical resection of liver metastases during the procedure of isolated hepatic perfusion. When inspecting the liver for potential biopsies, the surgeon has to take an educated decision on where to biopsy and how much to biopsy for it to be safe. There are significant differences in how large the biopsies are both because of tumor load and appearance of the tumors from the gross examination. Most often the biopsies are 2-3 mm in diameter and from these small biopsies we have wanted to do WGS/RNA-seq, PDX models and isolate TILs (for follow-up our studies of adoptive cell transfer published in Nature Communications 2017, Jespersen et al.⁸ and Figure for Reviewers, **Fig. R2**). Hence, we very early took the decision to generate young TIL cultures on parts of the biopsies and the reason for doing scRNA-seq on these as opposed to biopsies was a limitation of the material and that we wanted to capture as many TCR sequences as possible. All TIL cultures used for the scRNA-seq were cultured in a ten-fold lower concentration overnight before used in the scRNA-seq experiment as a means to reduce the impact of IL-2.

Nevertheless, we are mindful of that IL-2 will cause selective expansion of subclones of TILs and will impact gene expression despite that the levels were lowered before the experiment. We do not think we overinterpreted our data in our first version of the manuscript but we have altered the manuscript to tone down interpretations. We hope it will be kindly viewed upon that we have significantly revised this section and **Fig. 4**. It has also been reduced, and we mainly limit ourselves to conclusions that are justified based on data from non-expanded cells from biopsies. Much of the 10x-based analysis has been relegated to the supplementary material. We have not removed the latter entirely, since we still believe it has supporting value, given that the biopsy material in several cases contains very few cells (the reason we expanded them for the RNA analyses), in which case it can be useful to be able to compare expanded TILs to non-expanded TILs. However, the conclusions we draw based on the RNA-seq data from the expanded cells now mainly restrict themselves to the observation that T cell phenotypes are tied to clonotype. This is a conclusion we find little reason to believe would be in any way biased by IL2-driven expansion. Differences in gene expression-based clustering based on TCRs between different clonotypes ought to be due to pre-existing difference in phenotype. All CD8+ (and other cells) in a given sample has been subjected to the same IL2-treatment. While the exact profile of a given clone may be to some extent different after expansion, the fact that the clone

differs compared to other clones of the same cell type and in the same sample at all is unlikely to be caused by IL2. In addition, the RNA-seq data also gives us TCR sequences of T cells present in the samples, which can be useful in further studies to determine antigen-recognition.

In addition, we have a certain interest in knowing what T cells look like after expansion, as it is a part of the procedure for adoptive cell transfer (ACT). Presence of certain immune checkpoint receptors is relevant on expanded TILs in this context. We do demonstrate that TIL populations exist that are antigen-reactive, that PD-1+CD39+ subsets exist that indicate tumor recognition but that both before (biopsy material) and after expansion there is abundant expression of some checkpoint receptors, such as TIM-3, although TIGIT expression does appear more abundant in expanded TILs. This would be relevant to know about if expanded TILs are used for therapy.

We also have data on an *in vivo* experiment that demonstrates that expanded TILs from one of the tumors can eliminate its corresponding tumor in an immune-PDX model (Figure for Reviewers, **Fig. R2**). This is the first time our hIL2-NOG mouse model has been used in another diagnosis than cutaneous melanoma. We have not had the same success for all tumors, however, and the factors that determine whether TILs are suitably able to respond to the tumors or not remains to be resolved. But given that immunotherapies for UM have generally shown abysmal results, our data at least points to the conclusion that it is not only because the tumors are mainly surrounded by incompetent bystander T cells, but that other mechanisms are at play. This may include TIM-3, or other factors such as immune exclusion or tumor-intrinsic variables (possibly genes regulated by *BAP1*, alteration of which is both associated with higher levels of immune infiltration in poor-prognosis primary tumors⁹). Given our interest in immunotherapy and our novel immune-humanized mouse model we will use the information gained from this study in designing and testing novel approaches to tackle UM.

3. *The authors evaluate the functional impact of BAP1 loss on tumor phenotype by reintroducing the gene in a xenograft-derived cell line. While the findings are interesting, independent confirmation in at least one additional cell line would be desirable.*

We agree that this would be desirable, however, as mentioned above, the UM22 cell line we established here is the only *BAP1* mutated UM cell line among the very few that exist that we have been able to actually grow, and it will be contributed to the community. The cell line is a whole contribution in and of itself. On the other hand, we find compatible (reverse) regulation of markers previously investigated by Harbour et al.^{7,10} in knockdown experiments performed on the *BAP1* wild-type 92-1 cell line, which are compatible with the subtype-switch we describe. In addition, we find that 2358 of the significant genes also differ in a compatible manner between *BAP1* mutated and wild-type TCGA UM tumors (new **Fig. S5i-k**), supporting that a majority of the genes we find regulated overlap to a large extent to those that differ between subtypes that are characterized by *BAP1* mutation status. What we show, even with one cell line, is that *BAP1* can regulate this large set of genes. What we would gain with additional cell lines is to better define under which conditions it can regulate those genes and if other factors are involved as well in affecting the gene expression response to perturbation of *BAP1*. This would certainly be of interest, but we consider the conclusions we have drawn from this one experiment to be fully valid. See also the response to reviewer 1, comment 2 and reviewer 2, comment 1.

4. *It is also not clear how well the cell lines' transcriptome reflects that of the original tumor. The authors need to demonstrate that the cell line used is a meaningful representation of the in vivo situation.*

We do not believe that cell lines in general are accurate representations of the *in vivo* situation. The purpose here was to determine what happens transcriptomically to a *BAP1* mutated UM cell when *BAP1* is reintroduced. *In vivo*, tumor behavior and its transcriptome is affected by multiple factors in the tumor microenvironment, which in turn can affect the course of the disease. Here, we have taken interest in the function of one gene in terms of its effect on gene expression and the best way to get an interpretable readout of a genetic perturbation is in a pure cell line. UM22 is a *BAP1* mutated cell line that we have derived from one of the tumors in the study, which we argue is the closest representation of a metastatic UM we could get in cell line format, given that commercially available UM cell lines can be, on one hand, over a decade old, and, on the other hand are generally not *BAP1* mutated. No *BAP1* mutated cell line obtained by our lab has been able to grow sufficiently to perform any experiments on and we are uniquely enabled by this cell line to perform a reintroduction experiment. We have also heard from other scientists in the field that it is extremely difficult to establish *BAP1* mutant UM cell lines.

Regardless, we have compared this cell line to its parent tumor with respect to the genes we found regulated in the experiment and find that the cell line most closely correlates to its parent cell line (Figure for Reviewers, **Fig. R3a**). A more systematic comparison of all differentially expressed genes between the cell line and its parent tumor shows that the genes that differ have only 0.91% overlap with those we found regulated in the experiment, and none of these were any of the genes discussed in the study (**Fig. R3b**), showing that there is little concern that any differences in the biology of the parent cells and the cell line affect the response we find. Third, the genes that are differentially expressed between the cell line and the tumor are to a large extent enriched in two broad groups of genes: Immune-related genes (less in cell line), including CD8A and CD4, but notably not the ligand-related genes discussed in the main text, and cell cycle genes (more in cell line) (**Fig. R3c**). This is readily explained by the fact that immune-infiltration exists in the tumor, but not in the pure cell line and that cell lines growing in culture generally have higher cell cycle activity.

Furthermore, we compare the genes we found regulated in the experiment to differences between *BAP1* mutated and wild-type TCGA UMs and find that 2358 of them have compatible significant differences between these two subtypes, showing that the differences we observe in this cell line are differences that are relevant to those that distinguish *BAP1* mutated and wild-type tumor subtypes in patients (**Fig. S5i-k**).

5. *Tumor-infiltrating lymphocytes (TILs) are analyzed by flow cytometry (ex vivo and after expansion), as well as by TCR and RNA sequencing after expansion. The fact that TILs underwent a two week expansion in the presence of high-dose IL-2 precludes any meaningful interpretation of the single-cell sequencing analysis. While the authors claim that CD4/CD8 proportions are well-maintained, this is not true for at least half of the patients. Figure S9 further indicates that phenotypic changes will occur during culture.*

We agree that this can be a concern and have redone and reduced this entire section and **Fig. 4** as outlined in the response to reviewer 2, comment 2, but we disagree that no conclusions can be drawn from the study of such cells. We still consider data from expanded TILs to enable us to conclude that different clonotypes exist in different activation states within the same tumor. All CD8+ T cells in a given sample has been exposed to the same IL2 treatment and a systematic clustering of different CD8+ (and CD4+) T based on clonotype is unlikely to be caused by IL2. At most, there can be pre-existing differences in activation states that then also evolve differently after this stimulation, but the fact that they differ from each other remains. Beyond this, the remaining conclusions we draw are restricted to those that are motivated when considering non-expanded TILs from the biopsies, whereas the majority of the previous transcriptomics data on expanded TILs are kept for comparison in supplementary material (**Fig. S7a-d**).

6. *Analysis of T cell infiltrates in patients (cutaneous UM) that have recently undergone chemotherapy should be interpreted with caution, as lymphocytes are likely diminished as a result of cytotoxic therapy. The authors should comment on how/if this issue was addressed.*

One sample in the immune analysis had been subjected to previous chemotherapy: UM22. We agree that the proportions of different immune cells in this sample can be affected by the therapy and differ from the other samples for this reason. It is possible that this is one factor that could explain why, for instance, this sample has higher amounts of NK cells than any other sample. However, an equally plausible factor that could influence this difference is that this is the only skin metastasis of the eight profiled by single-cell methods, and metastases growing subcutaneously may be subjected to a different immune environment. It is also possible that some subclones within the tumor could have lost HLA expression and attracted these NK cells, while we do not detect HLA loss for the tumor as a whole. As described above, we have redone this section in relation to the other issues raised by the reviewer and we have limited our conclusions to the following, which we believe are valid even in perspective of the previous chemotherapy of this one sample: a) antigen-reactive T cells exist in this and other samples, b) PD-1+CD39+ T cells exist in this and other samples, as determined from biopsy TILs and c) different T cell clones exist in different gene expression states (determined from scRNA-seq). A global difference in the proportions of immune cells that are present in this particular sample, which could be influenced by previous chemotherapy, is not something we find likely to significantly alter these conclusions. Nonetheless, we have added to the description of this figure (revised **Fig. 4**) a statement that points out that UM22 can differ from the other samples as a consequence of the

previous chemotherapy.

7. *Dextramer staining is performed for a subset of patients. The kit used includes only HLA-A2 multimers, but no information on the selection and HLA-types of the analyzed patients is given. Data supporting HLA-A2 expression in the selected patients should be included.*

HLA typing was performed based on whole-genome sequencing data using the polysolver algorithm and, separately, on RNA-seq data using the OptiType algorithm, and the results have now been added as **Table S8**. We also show expression of the HLA-A gene (any genotype) in the tumors in **Fig S6d**. We performed dextramer staining experiments focusing on HLA-A2, on all samples (15) with this genotype, since MART-1 and gp100 are known to bind to this allele. The samples that are not in **Fig. 4b** were negative. This information has also been added to the figure legend. However, for the revised version, we have completed this panel with additional samples that we did not originally assay since they were part of a later batch that was obtained after the first dextramer staining experiment was performed. The previous samples included in the panel were also reanalyzed at the same time using a more powerful instrument for detecting binding events, yielding some minor differences in the proportions shown in **Fig. 4b**. While doing so, we also tested for reactivity towards NY-ESO-1, but found little to no reactivity towards this antigen (one sample showed weak reactivity). **Fig. 4b** has been updated with these results.

8. *The materials and methods section describes the definition of doublets from single cell data, however it is not clear how doublet events are handled. Optimally, these should be excluded from down-stream analysis, though in Figure 4d they appear to be listed alongside singlet events and analyzed as separate groups, though the meaning or significance of this is not made clear. In addition, please note that about 30% of T cells will express two alpha chains – this characteristic can thus not be used for doublet exclusion.*

We have redone doublet detection to allow for at most two different TCR alpha and/or beta chains as long as any of those extra chains are not also observed in any unique alpha/beta combination that exist in singlets. We have observed that in most cases with cells have multiple TCR chains detected, cells that have subsets of these chains in pairs of one alpha and one beta also exist and these generally have higher cell counts in the dataset, giving rise to the suspicion that most multiple alpha/beta-chain cells are likely to be doublets. Often these cells also clustered together separately, sometimes with intermediate CD4/CD8 phenotypes. While it might be possible that even more alpha or beta chains than we allow in this revised version could be expressed in a single T cell as well, realistically we find it more likely that they would be doublets, and we prefer to be on the side of caution. We have now also excluded all cells labeled as potential duplicates and have remade all analyses and figures related to the 10x data (**Fig. 4c**, **Fig. S7a-d**) to reflect this. Differences compared to the previous approach were minor.

9. *The plots depicted in Figure 4d suggest that large clonotypes significantly contribute to the clustering of the single cell data. This bias should be removed to better enable interpretation of the TIL phenotypes.*

Yes, this is one of the conclusions we draw and is the reason why we show both cell types and TCR-based labeling of clusters. It was of interest to us to see whether this could be the case. This is not something we consider a bias, but a source of variation that is of interest. Neither would it make sense to regress out the association between gene expression and clonotype here: First, this would preclude studying differences in activation states between clonotypes; Second, since CD4+ and CD8+ T cells predominantly have different clonotypes here, such a regression would also preclude distinguishing between the global expression patterns of CD8+ and CD4+ cells.

10. *The description of the correlation of clonotypes with cytotoxicity and exhaustion markers is difficult to follow, both in the materials and methods section and in the figure legend supporting Figure S8. The figure legend further states that the phenotype is derived from flow cytometry, rather than transcriptomics. Is this correct? If so, i) what is the exact phenotype used and ii) is the result supported by transcriptomics data?*

This analysis has now been redone in a simplified manner, instead showing the mean expression levels of the markers for each clonotype in a scatter plot (**Fig. S7d**). A plot on PD-1 vs CD39 expression was done in a similar manner (**Fig. 4c**). This is based on transcriptomics data, which enables detection of

clonotype sequences. Biopsy data on PD-1 vs CD39 expression can be seen in **Fig. 4a and S6c**. The levels of activation markers are shown for both the biopsy and transcriptomics data in **Fig. S7d-f**, and **S8**, as well as expression detected in bulk RNA-seq of tumors in **Fig. 4d**.

11. *Figure 4: the legend must reveal that analysis expanded TILs is shown*
The revised Fig. 4 uses TILs directly from biopsies in panels **4a** and expanded TILs in **4b-c**, and this has been stated in the revised legend.
12. *Figure 4: To allow better comparison between samples, Shannon's entropy is often normalized to the observed diversity.*
We have now calculated normalized entropy scores instead, using the formula in Kosman et al.¹¹ (dividing by log(N), where N is the number of unique clonotypes in a given sample). This panel is now in **Fig. S7c** (note that samples have been reordered according to name in this version).
13. *Figure S7: no legend exists for panel c) and d)*
The figure legend has been corrected with descriptions for panel c), which now in **Fig. 4a**, whereas panel d) has been removed, as it was deemed somewhat redundant.
14. *Figure S9: the number of (acquired) CD4 and CD8 cells within a single patient appears to be different between the plots showing PD1/TIGIT and FSC/CTLA4. Please comment.*
The flow cytometry plots in Fig. S9c-d (now **Fig. S8c-d**) were erroneously mixed up, causing CD4+ T cells to be displayed as CD8+ T cells and vice versa, this has now been corrected. Regarding the number of events shown in the plots, we have opted to show 100% of all events up to 10 000 events.
15. *Figure S10: no such figure was uploaded*
This figure was omitted by mistake and is included in this version (**Fig. S7e-f**).

Reviewer 3

1. *The data on transfection with functional BAP1 is really novel and very interesting, but is only performed in one sample.*
For the *BAP1* reintroduction experiment, we used UM22, since it is the only UM cell line that is *BAP1* mutated that we were able to establish and grow well enough to perform the experiment on. Other scientists in the field have also confirmed to us that it is extremely difficult to establish *BAP1* mutant UM cell lines. We have made a note and provided references in the Discussion. An inverse experiment, *BAP1* knockdown in a *BAP1* wild-type UM cell line (92-1), has been performed by Harbour et al. and we find consistent (reverse) changes in markers they investigated^{6,10}. See also the responses to reviewer 1, comment 2; reviewer 2, comment 1, and reviewer 2, comment 3..
2. *Similarly is the experiment on siRNA blockage of proliferation responsible genes very interesting, but again only in one selected sample.*
We have now performed knockdown-experiments in two additional cell lines, MP41 and 92-1. The results are shown in **Fig. 2h** and **Fig. S3c-d** and while the magnitude of the effects differ for some genes between cell lines, the overall pattern is similar. See also the responses to comments reviewer 1, comment 1 and reviewer 1, comment 6.
3. *In 8 samples are tumor infiltrating lymphocytes analyzed, but no clear conclusion from these experiments is given.*
In relation to comments from reviewer 2, we have redone this analysis and reduced it in scope (see revised **Fig. 4** and **Fig. S6-8**). We hope that the rewritten section also makes the conclusions more clear. The conclusions we draw are the following: We investigate whether lack of immunogenicity commonly observed in UM is likely to be due to tumors not containing reactive T cells, but rather only bystanders; a) Several of the profiled metastases contain antigen-reactive T cells as determined by dextramer binding assays; b) The T cell communities have PD-1+CD39+ subsets, which are markers that have been proposed to distinguish between tumor-reactive TILs as opposed to bystanders¹²; c) Individual TCR clonotypes are associated with different activation states, some of which have higher levels of these markers than others. In addition, we also gain a better understanding for the

phenotypes of TILs after expansion, as would be done prior to ACT. We have also performed an experiment that demonstrates that expanded TILs from one of these samples also can eliminate its corresponding tumor *in vivo* in PDXes, which serves to further demonstrate that this tumor does contain tumor-reactive TILs (Figure for Reviewers, **Fig. R2**). In all, this supports that lack of immune responses in these tumors are not primarily because the tumors do not contain tumor-specific TILs, but points to other mechanisms being active. In addition, we also find that it is possible to harvest TILs found in tumors, expand them and at least in some cases eliminate the tumor *in vivo*. See also the response to reviewer 2, comment 2.

4. *Setup: The authors compare metastases from liver and skin and from both anterior and posterior part of the eye. It can be questioned whether all parts of uvea are comparable. AJCC distinguishes between iris melanomas from the anterior part of the eye and choroidal/ciliary body melanomas from the posterior part of the eye. The authors do find discrepancies that confirms this clinically separation. From figure 1b it can be argued that metastases from the skin is not comparable to metastases from the liver. 4/6 metastases from the skin have no deletions in chr 3 compared to 0/21 metastases from posterior uveal melanoma and 1/1 metastasis from anterior uveal melanoma, indicating that posterior uveal melanoma has a "non-skin-like" type in contrast to skin and iris.*

Our goal was to describe and compare the alterations that occur in metastatic UMs, and contrasting heterogeneous tumors was of interest. In other words, the different types mentioned may not in all respects be identical but one goal here is to compare them and determine how they differ, for precisely this reason. Optimally, we would rather see even greater representation from tumors of different types, such as rare metastatic sites and primary sites to further investigate how they may differ from each other. Any observed differences between tumors that metastasize to different sites are definitely of interest here.

As for the differences observed, it does indeed appear that all cases with diploid chromosome 3 happen to be subcutaneous metastases, although the inverse is not the case. Namely, not all subcutaneous metastases lack LOH of 3. Taking into account LOH in UM22 (see response to reviewer 3, comment 11), this becomes 3/6 subcutaneous tumors that have LOH of 3. Furthermore, we also found *BAP1* mutations or deletions in all subcutaneous tumors, which is arguably the most relevant consequence of chromosome 3 alterations. Therefore, it is unclear whether, on one hand, if the difference is large enough to be meaningful, and, on the other, whether it is biologically relevant since *BAP1* is mutated in all subcutaneous tumors here.

Further, while all subcutaneous metastases were choroidal uveal melanomas, so were also at least 22/26 (two of unknown uveal site) of the liver metastases, making it difficult to conclude based on this data that the relatively greater frequency of chromosome 3 diploidy in the subcutaneous group is really due to the distinction between posterior and anterior tumors or a statistical coincidence. One could speculate that retaining both copies of chromosome 3 could contribute to an ability of metastasizing to the skin environment, or, alternatively, making it more difficult to metastasize to the liver. This speculation would be consistent with a previous report noting that among the few class I tumors (generally chromosome 3 diploid) that metastasize, a higher proportion choose other metastatic locations than the liver compared to class II tumors (which generally has monosomy of 3)¹³, suggesting that chromosome 3 status could be one factor involved.

The iris tumor displays a more notable difference in terms of mutations compared to the rest of the cohort, since it is much more highly mutated and has outlier high levels of PD-L1 expression. Had we excluded this tumor from the study on the basis of it being an anterior tumor, we would not have known that it has this distinct and highly relevant profile. Interestingly, the patient was also included into the PEMDAC trial (HDAC inhibitor entinostat and anti-PD1 antibody pembrolizumab) based on the high mutational burden and responded to the immunotherapy (manuscript in preparation).

5. *Why was all genetic experiments not performed on all samples? Could there be a bias introduced affecting the results?*

The number of samples included differed in 2 analyses: the mutational signature analysis in **Fig. 1e** and the study of gene-copy number-associations in **Fig. 2f**.

In the first case (**Fig. 1e**), this was because we thought it best to only include tumors with paired normal samples available to avoid any potential concern that possible population variants less well filtered out in tumor-only samples could influence the analysis. There are also two samples

expanded in PDXes that were excluded for this reason. However, we have revised the figure to include all samples in the analysis, after also performing a stricter filtering of variants to include (demanding minor allele read support > 10, in addition to the previous filtering). When including all samples in the panel after this (or before) there does not appear to be any indication that the signatures obtained from tumor-only (or the two PDX-derived samples) and tumor-normal-paired samples would differ as a result of the differences between these previously excluded samples. Although, taking into account all samples, we also found that signature S1 is abundant enough to be reported (as seen in the revised panel).

In **Fig. 2f** we could only analyze samples with paired RNA-seq data, which was available for 26/32 samples. In this revised version, we have also obtained matching RNA-seq data for 2 of the previously excluded tumors and have rerun the analyses in **Fig. 2f-g** to reflect this, with minor changes in the resulting candidate gene lists (**Fig. 2f**) but also some changes in enriched pathways (**Fig 2g, Table S4-5**). This is also the reason we have now split the set of genes in the siRNA experiment performed on the candidates in **Fig. 2h** to include the first-ranked candidates from **Fig. 2f-g** (which were the same as with the previous analysis) in the main panel and provide corresponding experiments for secondary candidates in **Fig. S3c** (where a few differed based on our criteria of choosing the secondary candidate per region as the highest ranked Cancer Gene Census gene).

6. *The preparation of cm3 + cm 4 and ocular specimens are not described in the paper.* CM3 and CM4 (now relabeled CM1 and CM2 for consistency), were processed identical to the UM samples and are one brain and subcutaneous metastasis of skin melanoma that are also described in a previous study¹⁴. This has been clarified in the Methods section. As a side note, the reason they were called CM3 and CM4 previously is due to two other skin melanomas (previously labeled CM1 and CM2) that were present in the same batch, which were not included here since they were PDXes (although, naturally, they also show a strong UV signature).
7. *Abstract: The authors states “..Lack of clinical rationale to biopsy metastatic disease”, which is clinically not correct as 9-18% of patients have metastatic disease from second primary cancer. This statement has now been removed.*
8. *The authors only investigate 2 matched primary tumors with their metastases so the statement “ .. some of which were not altered in matched primary eye tumors.” gives the impression that this match was done for more tumors. From figure 2C, which basically is the only information on this statement, there is a fairly good correlation between primary tumor and their metastasis and all clinically relevant changes can be found in both samples.*

We sequenced 8 matched primary tumors, but chose to highlight samples where we could be confident about differences, which included the copy number profiles of the two tumors in **Fig. 2c**. In **Fig. 2c**, chromosome 5 gain and 17p loss is present in the UM16 metastasis, but not the corresponding primary tumor. In UM25, loss of 6q is present in the metastasis by not the primary. This supports the findings in **Fig. 2b**, where 5p gain, 6q loss and 17p loss identified are significantly overrepresented in our metastatic cohort compared to unmatched primary tumors from TCGA. **Fig 2b** therefore suggested that these could be late events, which is then further shown by these differing between matched primaries and metastases in **Fig. 2c**. Underrepresentation and late acquisition of 6q loss and 17p loss therefore suggests that they may be associated with metastasis and there exists previous studies stating that 6q is associated with greater risk of distant metastasis¹⁵. These differences can therefore be clinically relevant, although the exact consequences of the events remain to be fully determined.

To further expand on similarities and differences between primary tumors and metastases, we have added a comparison of mutations identified in the matched pairs to Table S2. However, it should be noted that the primary tumors included in this comparison are obtained from FFPE specimens and have worse sequence quality, making it different to draw firm conclusions regarding individual mutations that differ (the copy number profiles, on the other hand, are binned averages across entire chromosomes and this still enables identification of clear gain/loss events at the chromosome arm level in low-coverage samples¹⁶). As another comparison, we have also compiled the relative mutation rates of all genes in our cohort compared to TCGA UM tumors (which are all primary) in **Fig. S1f-g** (mutations not shown there indicate that they are not shared between the two datasets).

9. *Introduction: The citation for PRAME does only state on the use for prognostication for Class 1 tumors, as class 2 tumors always have poor prognosis. So the binary separation is only for class 1 tumors. The reason we comment on PRAME expression for this tumor is that it lacks both BAP1, SF3B1 and EIF1AX mutations, which is rare in metastatic cases. This suggested to us that it could be a rare metastatic Class I tumor, as opposed to the other ones in our cohort, which have genomic alterations characteristic of Class II tumors. Therefore, elevated PRAME expression in this tumor was of interest, since it would be compatible with the subset within class I that is more prone to metastasis than the majority of Class I tumors¹³. The statement on page 4 has been expanded to clarify our reasoning and PRAME expression has also been added to Fig. S1e.*
10. *Results: P4: Did the authors look for functional BAP1 with IHC?*
We have performed IHC on the BAP1 reintroduced and BAP1 mutated (unperturbed) UM22 cell lines as well as a number of PDX tumors, which we have added in Fig. 3b-c and Fig. S4b-c. Our data shows that tumors expressing wildtype BAP1 exhibits nuclear BAP1 staining, whereas the tumors with BAP1 mutations either exhibit no BAP1 staining or cytoplasmic staining. This is now mentioned in the text.
11. *P4: UM28 had polyploidy, could the homozygotic loss of BAP1 be due to isodisomy?*
The tumor that was inferred to be polyploid, UM28, was the only sample without any discovered mutations in either BAP1 or SF3B1. It is likely that the reviewer instead refers to UM22, which we mention has loss of BAP1 via a homozygotic frameshift deletion in a setting where chromosome 3 appears to be diploid. When we inspect the allele frequencies of heterozygous germline SNVs on this chromosome, there is a notable shift in these frequencies on all of chromosome 3 towards homozygosity, despite the chromosome being diploid. This indicates that this tumor in fact has copy number neutral loss of heterozygosity or isodisomy of chromosome 3, also explaining why identical frameshift deletions in BAP1 occur on both alleles (duplication of the mutated chromosome followed by loss of non-mutated chromosome). Copy number neutral LOH also seems to be present on chromosome 2 in this sample, which also otherwise appears diploid (new Fig. S1b). We have updated the main text to describe that copy number neutral LOH occurs at chromosome 3 in this sample.
12. *P4: Again the use citation only states om PRAME and class 1 tumors.*
See the response to reviewer 3, comment 9 above. We were interested in whether this metastatic BAP1 and SF3B1-wildtype tumor could potentially be a rare poor prognosis class I type of tumor.
13. *P6: Comparison between TCGA material and the metastases from this material seems not appropriate as the TCGA contained data from both metastatic and non-metastatic tumors. TCGA showed a significant difference between the metastatic and non-metastatic groups. Therefore all findings would automatically be significant, due to the basic difference (Figure 2b)*
TCGA UM tumors are all primary. While they refer to some of them as metastatic, this is used to indicate that the patients have later developed metastatic disease, whereas the tumors sequenced are not themselves metastases. Correlation tests were not performed in a combined dataset of TCGA and our data, but rather in each separately. The ones common in both were then considered potential candidates.

Nonetheless, in the text we mention that the obtained gene lists were enriched for genes associated with the Class I/II subtypes. What may raise the reviewer's concern is that one may therefore expect that the expression of genes in regions affected by subtype-associated copy number changes could differ between the subtypes in the TCGA dataset due to either the copy number changes themselves or due to other factors that differ between the subtypes (such as BAP1 status). We are able to distinguish between genes that differ due to the basic difference between subtypes and those that differ due to dosage effects of copy number changes by intersecting the genes found correlated in TCGA with those that are also correlated independently with copy number within our entirely metastatic dataset, where also nearly all tumors are BAP1 mutated (among the three wild-type tumors in our dataset, only two had RNA available that were included in the analysis, which are unlikely have any major influence). Therefore, the genes in the intersect associate independently with copy number changes themselves. As an example, if we begin in the other end and only consider the genes found correlated with copy number in our dataset, without comparing to TCGA, that list contains all the same genes presented here, except also a number of others that we then lose when demanding that they are consistent also in TCGA.

There is one type of bias worth pointing out here, however and that is that intersecting the two studies does not enable the study of genes that are altered by some copy number changes that only, or mostly occur in class I tumors, given that our dataset is almost exclusively representative of class II. This can explain enrichment of class II-associated genes, but such enrichment would not occur if the genes themselves were not also consistently regulated by dosage effects of copy number changes across both datasets. We can still conclude from the enrichment test that it further supports that these class II-associated genes are affected in terms of expression by these copy number changes, and that this therefore also shows that the copy number changes contribute (together with other factors, perhaps mainly *BAP1* mutation) to forming the transcriptome of this subtype.

In summary, we see no reason to believe that both good and poor-prognosis tumors being present in the TCGA data would in any way would give automatic significance to the individual genes in the final list, especially considering the vast differences between these datasets: with the TCGA dataset being from a separate batch consisting of primary tumors of unrelated patients, and the fact that the correlations exist independently within our metastasis-only dataset. The common denominator between these distinct datasets that can explain regulation of these genes is the copy number changes.

14. *P6: The mentioned survival, is that given from the HPRD database or is it the observed survival?*
This refers to observed survival as defined in TCGA clinical metadata (days to death) of these samples. The text now states “association with observed survival” (page 6) to clarify this. The methods section also describes that the clinical metadata is downloaded from TCGA.
15. *P6: the siRNA are really interesting, but only one, highly different from the other samples was used, and therefore conclusions should be made very cautiously.*
As described in the response to reviewer 3, comment 2, we have now performed knockdown-experiments in two additional cell lines, MP41 and 92-1. The results are shown in **Fig. 2h** and **Fig. S3c-d**. We feel that the conclusion “Thus, these recurrent arm-level copy number changes contribute to shaping the transcriptomic subtypes of UM and regulate genes that may conceivably contribute a fitness advantage” is sufficiently cautiously phrased.
16. *P7: were the transfected cells stained for IHC BAP1 to show functional BAP1?*
As described in the response to reviewer 3, comment 10, we have performed IHC on the *BAP1* reintroduced and *BAP1* mutated (unperturbed) UM22 cell lines (new **Fig. 3b-c**), as well as PDXes established from UM25, UM22, UM1, UM6, UM7, UM9, UM23, UM24 (new **Fig. S4b-c**). This show lack nuclear staining for *BAP1* in the mutated cell line and full staining in the *BAP1*-reintroduced cell line.
17. *Discussion: P9: biopsy of metastasis is meaningful – see above*
See the response to reviewer 3, comment 7.
18. *P9: Comparison of TCGA data with this dataset – see above*
See the response to reviewer 3, comment 13.
19. *Figure 1a: would improved if number of samples were given for each sub-experiment*
The number of samples have now been indicated in the panel and clarified in the legend to **Fig. 1a, 1e** and **Fig. 2f**. See also the response to comment reviewer 3, comment 5.
20. *Figure 1e: seems some information is missing (cm3, cm4, um25, um28, um29, um30 and um31).*
The missing metadata has been added for these samples, although the category “Part of uvea” is not applicable to the two cutaneous melanomas.
21. *Why was um4, um5, um6, um15, um16, um22, um23, um25 and um31 not analyzed?*
See the response to reviewer 3, comment 5 for the reasoning behind that choice. We have, however, included all samples in this revised version of **Fig. 1e**.
22. *Figure 2a: um19 is stated in figure 1b to have large deletion in chr.3 but in this figure it seem chr.3 is normal. Is the low purity (figure S2) causing misinterpretation of this sample?*
The loss of chromosome 3 in UM19 is only weakly visible in **Fig. 2a**. The color intensity reflects copy

number relative to a diploid expectation, but if the fraction of diploid non-tumor cells in a sample is large, the color therefore becomes weaker. As seen in **Fig. S1d**, the purity of this sample was only estimated to about 10% and this is very likely the reason for the low estimated copy number amplitude and weak color seen in **Fig. 2a**. We could, alternatively, normalize the observed copy number amplitude to tumor purity, but we reasoned that avoiding doing simultaneously visualizes both differences in copy number events (in the binary sense of whether they are present or absent) as well as tumor purity (although the latter is shown separately in **Fig. S1d**). For the secondary analysis of these copy number changes with respect to effects on gene expression (**Fig. 2f**), we include tumor purity in the linear model that determines association between gene expression and copy number to account for these purity differences.

23. *Figure 2f: What does the inner and outer circle represent?*
The inner circle represents summarized copy number profiles of TCGA tumors, and the outer circle our cohort. Explanatory text has been added to the panel to indicate this. This is also described the figure legend.
24. *Figure S3: the labelling of panels is not unambiguous.*
The panel labels and the corresponding legend have now been corrected. This is now **Fig. S2**.
25. *Figure S5: Is the panels showing IHC stains from 2 PDX experiments or 2 different area in the PDX or the patient metastasis and PDX?*
The panels show patient tumors and PDX tumors. This information has been added to the figure and the corresponding legend.
26. *Figure S9: c+d panel no explained in test*
Descriptions for panel c and d have now been added to the figure legend. This is now Fig. S8.
27. *Figure S10: c+d panel is nonexisting*
The figure legend incorrectly referred to panels c and d, and the legend has now been updated to remove this. These panels have been moved to **Fig. S7e-f**.

Figures

Fig. R1

Figure R1: Differential expression of checkpoint receptors and T cell markers between *BAP1* mutated and wild-type TCGA UM samples. All expressed genes were tested using rank-sum tests and *p*-values were adjusted with Benjamini-Hochberg correction.

Fig. R2

Figure R2: Patient-derived xenograft version 2 (PDXv2) experiment using matched UM1 tumor and TILs. Tumor cells were labelled in short-term culture with a virus expressing firefly luciferase and then transplanted into immunocompromised NOD/SCID IL2 receptor g knockout mice (NOG mice). When a tumor had developed this was divided into 2x2 mm pieces that were transplanted into five NOG mice or five NOG mice expressing human IL-2 (hIL2-NOG). Tumor growth was monitored by injecting luciferin into the mice and by imaging in a IVIS Lumina III XR. When luciferase measurements had increased two weeks in a row, 20 million TILs originally expanded from UM1 were injected into the hIL2-NOG mice. Tumor growth was followed until the tumor size in the NOG mice had reached ethics limit (10 mm of the smallest side of the tumor).

Fig. R3

Figure R3: a) Correlations to the UM22 cell line (unperturbed) on the basis of the genes identified as differentially expressed in the BAP1 experiment. Spearman correlation coefficients were calculated on the basis of \log_2 (RPKM+1) normalized and batch corrected values, on the basis of all genes differentially expressed in the BAP1-reintroduction cell line experiment at $q < 0.05$ and absolute \log_2 fold change > 1 . b) Venn diagram comparing genes identified as differentially expressed in the BAP1 experiment and genes differentially expressed between cell line and tumor biopsy ($q < 0.05$, compatible direction of fold change). Differential expression tests were carried out with DESeq2, taking into account sample batch. c) Enriched MSigDB pathways among genes differing between UM22 cell line and tumor, with respect to the MSigDB “canonical pathways” category.

References

1. Feng, X. *et al.* A Platform of Synthetic Lethal Gene Interaction Networks Reveals that the GNAQ Uveal Melanoma Oncogene Controls the Hippo Pathway through FAK. *Cancer Cell* **35**, 457-472.e5 (2019).

2. Liu, J. *et al.* An Integrated TCGA Pan-Cancer Clinical Data Resource to Drive High-Quality Survival Outcome Analytics. *Cell* **173**, 400-416.e11 (2018).
3. Van Raamsdonk, C. D. *et al.* Frequent somatic mutations of GNAQ in uveal melanoma and blue naevi. *Nature* **457**, 599–602 (2009).
4. Amirouchene-Angelozzi, N. *et al.* Establishment of novel cell lines recapitulating the genetic landscape of uveal melanoma and preclinical validation of mTOR as a therapeutic target. *Mol. Oncol.* **8**, 1508–1520 (2014).
5. Bononi, A. *et al.* BAP1 regulates IP3R3-mediated Ca²⁺ flux to mitochondria suppressing cell transformation. *Nature* **546**, 549–553 (2017).
6. Harbour, J. W. *et al.* Frequent Mutation of BAP1 in Metastasizing Uveal Melanomas. *Science* (80-.). **330**, 1410–1413 (2010).
7. Field, M. G. *et al.* BAP1 loss is associated with DNA methylomic repatterning in highly aggressive class 2 uveal melanomas. *Clin. Cancer Res.* **25**, 5663–5673 (2019).
8. Jespersen, H. *et al.* Clinical responses to adoptive T-cell transfer can be modeled in an autologous immune-humanized mouse model. *Nat. Commun.* **8**, (2017).
9. Robertson, A. G. *et al.* Integrative Analysis Identifies Four Molecular and Clinical Subsets in Uveal Melanoma. *Cancer Cell* **32**, (2017).
10. Matatall, K. a *et al.* BAP1 deficiency causes loss of melanocytic cell identity in uveal melanoma. *BMC Cancer* **13**, 1–12 (2013).
11. Kosman, E. & Leonard, K. J. Conceptual analysis of methods applied to assessment of diversity within and distance between populations with asexual or mixed mode of reproduction. *New Phytol.* **174**, 683–696 (2007).
12. Simoni, Y. *et al.* Bystander CD8+T cells are abundant and phenotypically distinct in human tumour infiltrates. *Nature* **557**, 575–579 (2018).
13. Field, M. G. *et al.* PRAME as an Independent Biomarker for Metastasis in Uveal Melanoma. *Clin. Cancer Res.* **22**, 1234–42 (2016).
14. Bagge, R. O. *et al.* Mutational Signature and Transcriptomic Classification Analyses as the Decisive Diagnostic Tools for a Cancer of Unknown Primary. *JCO Precis. Oncol.* 1–25 (2018). doi:10.1200/PO.18.00002
15. Benjamin A Krantz, Nikita Dave, Kimberly M Komatsubara, B. P. M. & Carvajal, R. D. Uveal melanoma: epidemiology, etiology, and treatment of primary disease. *Clin. Ophthalmol.* **11**, 279–289 (2017).
16. Adalsteinsson, V. A. *et al.* Scalable whole-exome sequencing of cell-free DNA reveals high concordance with metastatic tumors. *Nat. Commun.* **8**, (2017).

Reviewers' Comments:

Reviewer #1:

Remarks to the Author:

Most of my previous points have been addressed but some remain unclear to me.

2. BAP1 regulation of targets at the protein level

Fig. S5e is simply a Venn diagram. In tumor sections, is there a decrease in expression by IHC in any of the targets? Or in mutant BAP1 level lines re-expressing wild-type BAP1?

7. I am asking whether the level of BAP1 is physiological i.e. how does it compare to wild-type BAP1 re-expressing cells.

Other points

1. I would not describe FAK as a negative regulator of cell detachment-initiated apoptosis. FAK is a focal adhesion associated protein that is tyrosine phosphorylated following integrin engagement and de-phosphorylated followed detachment.

2. A few typos: page 7 line 211 "wildtype"

Reviewer #2:

Remarks to the Author:

With respect to the relevance of analyzing expanded TILs, I continue to disagree with the authors, but concede to their point of view as expressed in the current version of the manuscript.

For the association between clonotype and transcriptome, not removing/regressing TCR genes will impact PCA/cluster formation, in particular where large clones are involved.

Unfortunately, Supplemental Figure 7, which should illustrate tSNE of phenotypes and their overlay with clontypes, opens up to display Supplemental Figure 8, so it is not possible to further judge this issue and the data of Figure 4C.

To remove bias or demonstrate that no bias exists, the authors need to perform their phenotypic analysis without allowing contribution of TCR genes. Annotation of clontypes will still be possible and the authors should give examples of location of clontypes highlighted in Figure 4C in tSNE plot in conjunction with their expression of PD1 and CD39.

Reviewer #3:

Remarks to the Author:

The revision of the paper "Molecular profiling of driver events in metastatic uveal melanoma" by Karlsson et al. has overall improved the manuscript. The comments from the reviewers have been answered satisfactory. I do, however, still have some reservations concerning the paper

Introduction:

L 56: The statement "Uveal melanoma (UM) is a rare form of melanoma but the most common intraocular cancer" is not correct as retinoblastoma worldwide is more common than intraocular melanoma. But in adult Caucasians uveal melanoma is the most common intraocular tumor. The reference used is not supporting the statement. The used reference is also misciting the original reference.

L 58: "... generally..." no good evidence that metastatic disease can be cured exists, and generally should be removed. The used reference should be changed to Rantala et al. 2019 1. A recent paper on combined immunotherapy and promising results do exists, however it is a retrospective study2.

L66: "...in a close to binary manner" Even though GEP seems to be superior to other prognostication models increasing evidence suggests that separation in two groups is not possible³⁻⁵

L70 The reference should be changed to Rantala et al 2019 1

L71-73: information really not relevant for the message of the paper

Results:

L86: "two cases were ambiguous" why was that? Later in figure 1 the cases are stated as not known. Were the location not known or were the tumors overlapping different parts of uvea?

L114: In the third tumor (UM28) without BAP1 mutation, did the authors look for functional BAP1 protein? This case has apparently no driver mutation, and epigenetically BAP1 could have been turned off (see shain et al case A13 and A29 6).

L159+Figure 1b+Figure2a+b: The data regarding loss of chromosome 3 is ambiguous as in table 1b 1 case with LOH of BAP1 and 4 cases with no loss of CHR3 =5, in Figure 2A 5 tumors have not lost CHR3, but in Figure 2b it is stated that 6 cases have disomy 3. Also, the iris melanoma is included in this number, whereas the TCGA excluded iris melanomas because of the question whether this type of melanoma can be compared to the posterior types of uveal melanoma. The authors do present data that proves exactly that anterior and posterior melanomas might have different pathogenesis.

Discussion:

L 284: The study by shain et al 6 Which is the largest study on matched primary and metastatic uveal melanoma is only referenced as an inferior study due to targeted sequencing, in the introduction. As the authors mainly find the same alterations in the metastases and only have few primaries, I think the authors should discuss this paper in relation to their own findings.

L293: If the authors did not investigate if functional BAP1 protein, the statement "we observed BAP1 alterations in 91%" should be changed to "we observed BAP1 mutations in 91% of".

L299: UV damage of UM have been reported by Royer-Bertrand et al.⁷ and Van Poppelen et al. ⁸ and they find opposite findings than reported in this study.

L312: CDKN2A alterations have been reported by Shain et al.⁶

1. Rantala, E. S., Hernberg, M. & Kivelä, T. T. Overall survival after treatment for metastatic uveal melanoma: a systematic review and meta-analysis. *Melanoma Res.* 29, 561–568 (2019).
2. Bol, K. F. et al. Real-World Impact of Immune Checkpoint Inhibitors in Metastatic Uveal Melanoma. *Cancers (Basel)*. 11, (2019).
3. Corrêa, Z. M. & Augsburger, J. J. Independent Prognostic Significance of Gene Expression Profile Class and Largest Basal Diameter of Posterior Uveal Melanomas. *Am. J. Ophthalmol.* 162, 20–27.e1 (2016).
4. Walter, S. D. et al. Prognostic implications of tumor diameter in association with gene expression profile for uveal melanoma. *JAMA Ophthalmol.* 134, 734–740 (2016).
5. Cai, L. et al. Gene Expression Profiling and PRAME Status Versus Tumor-Node-Metastasis Staging for Prognostication in Uveal Melanoma. *Am. J. Ophthalmol.* 195, 154–160 (2018).
6. Shain, A. H. et al. The genetic evolution of metastatic uveal melanoma. *Nat. Genet.* 51, 1123–1130 (2019).
7. Royer-Bertrand, B. et al. Comprehensive Genetic Landscape of Uveal Melanoma by Whole-Genome Sequencing. *Am. J. Hum. Genet.* 99, 1190–1198 (2016).
8. van Poppelen, N. M. et al. Genetic Background of Iris Melanomas and Iris Melanocytic Tumors of Uncertain Malignant Potential. *Ophthalmology* 125, 904–912 (2018).

Responses

Reviewer 1

2. *BAP1 regulation of targets at the protein level*
Fig. S5e is simply a Venn diagram. In tumor sections, is there a decrease in expression by IHC in any of the targets? Or in mutant BAP1 level lines re--expressing wild-type BAP1?

Fig. S5g also shows boxplots of protein expression levels of the genes discussed, where the decrease can be seen at protein level. The total proteomics data was also used in **Fig. S5f** for gene-set enrichment analysis with respect to the class I/II categories at the protein level. Besides this, we have also added a supplementary table (**Table S8**) with the abundances of all detected proteins. Selecting a few targets for IHC validation is not meaningful since it is more biased than the data that we have provided.

7. *I am asking whether the level of BAP1 is physiological i.e. how does it compare to wild-type BAP1 re-expressing cells.*

IHC for BAP1 in UM22 (**Fig. 3c**) after reintroduction can be compared to IHC for BAP1 in the wildtype tumors UM25 (**Fig. S4c**) and UM28 (**Fig. S4d**), the latter of which we have added in this revision, as well as the BAP1 wildtype cell lines 92-1 and MEL202 in (**Fig. S4b**). In addition, we have also redone the previous western blot and included the BAP1 wildtype cell lines MP41 and 92-1 for comparison in **Fig. 3b**. Values appear somewhat higher for the reintroduced UM22 cells compared to the other wildtype cells, but not excessively so. We also compared the expression of BAP1 mRNA in the BAP1 reintroduced UM22 cells and compared them to the expression in all samples. It shows that that the level is higher than most tumors but not all, suggesting that the level is in a physiological range.

This analysis shows that that the level is higher than most tumors but not all, suggesting that the level is in a physiological range.

1. *I would not describe FAK as a negative regulator of cell detachment-initiated apoptosis. FAK is a focal adhesion associated protein that is tyrosine phosphorylated following integrin engagement and de-phosphorylated followed detachment.*

As we understand it, phosphorylation upon integrin engagement promotes transduction of signals

that suppress an apoptotic program that would otherwise occur following cell detachment (anoikis)¹. De-phosphorylation following detachment instead allows for this apoptotic process to occur. The reviewer is correct that the negative regulation of this process refers to when cells attach, but not necessarily when they detach. In this sense, we admit that the previous phrasing was unclear. However, attachment-independent suppression of anoikis has also been proposed for experimentally induced activation of FAK² and via signaling from endosomal integrins³. In UM, FAK has also been proposed to be activated by mutated GNAQ/GNA11, although it is unclear if this activation influences anoikis⁴. We have rephrased the sentence to clarify this point. We now state “[...] a focal adhesion associated kinase known for being activated upon matrix-integrin interactions and thereby mediating survival signals that prevent detachment-associated apoptosis (anoikis)”. We have also replaced the previous review reference with primary references on this topic^{1-3,5}.

4. 2. A few typos: page 7 line 211 "wildtype"

We thank the reviewer for finding this typo. This typo and a few others that we could discover have now been corrected.

Reviewer 2

1. *For the association between clonotype and transcriptome, not removing/regressing TCR genes will impact PCA/cluster formation, in particular where large clones are involved. Unfortunately, Supplemental Figure 7, which should illustrate tSNE of phenotypes and their overlay with clonotypes, opens up to display Supplemental Figure 8, so it is not possible to further judge this issue and the data of Figure 4C. To remove bias or demonstrate that no bias exists, the authors need to perform their phenotypic analysis without allowing contribution of TCR genes. Annotation of clonotypes will still be possible and the authors should give examples of location of clonotypes highlighted in Figure 4C in tSNE plot in conjunction with their expression of PD1 and CD39.*

We have redone all analyses related to the single-cell RNA-seq data after removing all TCR genes from the dataset (about 232 genes removed) and updated all plots accordingly. The scatter plots in Fig. 4C are identical afterwards, since the TCR genes were not included in those calculations. The tSNE plots only show minor differences. We have confirmed that both Fig. S7 and Fig. S8 were actually uploaded last time. But Fig. S7 has been revised to also include tSNE plots that show the location of the clonotypes displayed in Fig. 4c (see Fig. S7e), and the added panel Fig. S8e similarly indicates clonotypes of relative to those shown in the scatterplots in Fig. S8a (previously Fig. S7d). The new Fig. S7d shows the expression of *PDCD1* and *ENTPD1* (CD39) at the single-cell level in tSNE plots that may be compared to the locations of clonotypes in Fig. S7e.

Reviewer 3

Introduction:

1. L 56: *The statement “Uveal melanoma (UM) is a rare form of melanoma but the most common intraocular cancer” is not correct as retinoblastoma worldwide is more common than intraocular melanoma. But in adult Caucasians uveal melanoma is the most common intraocular tumor. The reference used is not supporting the statement. The used reference is also misciting the original reference.*

The statement has been changed to “most common intraocular cancer in adults”, which should be correct since very few cases of adult onset retinoblastoma have been described⁶. The previously used reference has also been replaced with [7-9] for UM incidence and [6] for adult retinoblastoma incidence (since none of the UM-specific articles on incidence make direct comparisons with the latter).

2. L 58: *“... generally...” no good evidence that metastatic disease can be cured exists, and generally should be removed. The used reference should be changed to Rantala et al. 2019 1. A recent paper on combined immunotherapy and promising results do exist, however it is a retrospective study[2].*

The word “generally” has now been removed and the proposed reference has been added.

3. L66: “...in a close to binary manner” Even though GEP seems to be superior to other prognostication models increasing evidence suggests that separation in two groups is not possible[3–5]

It is true that this is a simplification. Additional subgroups can also be determined both based on molecular clustering as well as with specific clinico-pathologic parameters. We have rephrased the sentence and cited references on this, including the ones provided by the reviewer. Still, it is also the case that broadly speaking, UM largely falls into two groups, one of which is associated with BAP1 mutations and which has a distinct gene expression profile.

4. L70 The reference should be changed to Rantala et al 2019 [1]

The reference has now been changed to the one proposed by the reviewer.

5. L71-73: information really not relevant for the message of the paper

We have removed the sentence at L72-73, but retained the sentence at L71 to place into context how the biopsies were obtained.

Results:

6. L86: “two cases were ambiguous” why was that? Later in figure 1 the cases are stated as not known. Were the location not known or were the tumors overlapping different parts of uvea?

These two samples did not have information about exact uveal location of the primary tumor available in our journals, since the primary tumors were treated at another hospital. We have rephrased the sentence to clarify this.

7. L114: In the third tumor (UM28) without BAP1 mutation, did the authors look for functional BAP1 protein? This case has apparently no driver mutation, and epigenetically BAP1 could have been turned off (see shain et al case A13 and A29 [6]).

This tumor is very interesting since it has monosomy of 3 and has metastasized, but shows no sign of any BAP1 mutation. We have now performed IHC on this tumor, which shows that BAP1 protein is expressed and that it localizes to the nucleus, as is typical for wildtype tumors. It could still be possible that some complex mutation exists that evades detection, but for all we can tell it appears wildtype. One could speculate that loss of one copy of chromosome 3 alone either has some effect on tumor behavior or that unknown UM tumor suppressors inactivated by LOH could exist on this chromosome.

On a related note, we have revised **Fig. 1b** to also include a GNAQ Q209P mutation for UM20, which we have since discovered is present on RNA (added **Fig. S1b**), but which has poor read support on DNA.

8. L159+Figure 1b+Figure2a+b: The data regarding loss of chromosome 3 is ambiguous as in table 1b 1 case with LOH of BAP1 and 4 cases with no loss of CHR3 =5, in Figure 2A 5 tumors have not lost CHR3, but in Figure 2b it is stated that 6 cases have disomy 3. Also, the iris melanoma is included in this number, whereas the TCGA excluded iris melanomas because of the question whether this type of melanoma can be compared to the posterior types of uveal melanoma. The authors do present data that proves exactly that anterior and posterior melanomas might have different pathogenesis.

We thank the reviewer for finding this typo. **Fig. 2b** was incorrect here, it should be 5 cases without loss of chr. 3. We have fixed and updated those results, which leads to significance also for loss of 3. If the iris tumor is excluded from this analysis, the same chromosomal alterations remain as significant (17p loss, 8q gain, 5p gain, 6q loss and loss of 3), which is reasonable since it is only one tumor and its copy number profile is relatively similar to the others, besides lacking loss of 3. We still do not believe it is necessary to exclude it, but have added those results as well to **Table S3**. We also confirmed that the results in **Fig. 2f** were similar after excluding this tumor:

As for why TCGA may have excluded iris tumors, we do not see any information about this in their paper, so we can only speculate that it could be because iris tumors have a very low incidence and that they might have wanted a decent number of samples of each type when they performed consensus clustering to define molecular subtypes. The goal of our study was not to perform clustering and define new subtypes, so this is not a concern here.

Discussion:

9. L 284: *The study by shain et al 6 Which is the largest study on matched primary and metastatic uveal melanoma is only referenced as an inferior study due to targeted sequencing, in the introduction. As the authors mainly find the same alterations in the metastases and only have few primaries, I think the authors should discuss this paper in relation to their own findings.*

We have added discussion on some of our results in relation to Shain et al. in two places of the Discussion, including in relation to *CDKN2A* deletions as mentioned in the response to comment 12, reviewer 3.

10. L293: *If the authors did not investigate if functional BAP1 protein, the statement “we observed BAP1 alterations in 91%” should be changed to “we observed BAP1 mutations in 91% of”.*

The statement has been changed accordingly.

11. L299: *UV damage of UM have been reported by Royer-Bertrand et al.7 and Van Poppelen et al. 8 and they find opposite findings than reported in this study.*

Royer-Bertrand et al. (whom we have cited) performed mutational signature analysis of choroidal, ciliary and ciliochoroidal tumors and reported an absence of UV signatures for all samples. This is compatible with our study, where such signatures were also absent in all choroidal and ciliary tumors. We find evidence of UV damage only in iris tumors, which explains why Royer-Bertrand et al. did not report UV damage, since they did not study any iris melanoma.

Van Poppelen et al. performed targeted sequencing of only the genes *GNAQ*, *GNA11*, *EIF1AX*,

SF3B1, *BAP1*, *NRAS*, *BRAF*, *PTEN*, *c-Kit* and *TP53* in iris melanoma. They did not perform any genome/exome-wide mutational signature analysis as done by Royer-Bertrand et al. and us (in the manner defined by Alexandrov et al.¹⁰). In the discussion the authors mention that the variants they found in some of the cutaneous melanoma-associated genes, *NRAS*, *BRAF*, *PTEN*, *c-KIT*, and *TP53*, were not of the C>T or CC>TT types associated with UV damage. This is, however, far from sufficient to state that such a signature does not exist in the broader context of the genome. For example, even in cutaneous melanoma, the very common *BRAF* V600E mutation is not a C>T substitution. To determine whether a UV signature is present, one has to perform a genome, exome-wide or a very large targeted panel analysis and take into account the trinucleotide context of the substitutions¹⁰.

12. L312: *CDKN2A* alterations have been reported by Shain et al. [6]

We have added a mention of the observation by Shain et al. that *CDKN2A* loss can occur during metastatic progression and discuss this in the context of our observations.

References

1. Ilić, D. et al. Extracellular matrix survival signals transduced by focal adhesion kinase suppress p53-mediated apoptosis. *J. Cell Biol.* **143**, 547–560 (1998).
2. Frisch, S. M., Vuori, K., Ruoslahti, E. & Chan-Hui, P. Y. Control of adhesion-dependent cell survival by focal adhesion kinase. *J. Cell Biol.* **134**, 793–799 (1996).
3. Alanko, J. et al. Integrin endosomal signalling suppresses anoikis. *Nat. Cell Biol.* **17**, 1412–1421 (2015).
4. Feng, X. et al. A Platform of Synthetic Lethal Gene Interaction Networks Reveals that the GNAQ Uveal Melanoma Oncogene Controls the Hippo Pathway through FAK. *Cancer Cell* **35**, 457–472.e5 (2019).
5. Zheng, Y. et al. Protein tyrosine kinase 6 protects cells from anoikis by directly phosphorylating focal adhesion kinase and activating AKT. *Oncogene* **32**, 4304–4312 (2013).
6. Kaliki, S. et al. Newly diagnosed active retinoblastoma in adults. *Retina* **35**, 2483–2488 (2015).
7. Singh, A. D., Turell, M. E. & Topham, A. K. Uveal melanoma: Trends in incidence, treatment, and survival. *Ophthalmology* **118**, 1881–1885 (2011).
8. Virgili, G. et al. Incidence of Uveal Melanoma in Europe. *Ophthalmology* **114**, (2007).
9. Park, S. J. et al. Nationwide incidence of ocular melanoma in South Korea by using the national cancer registry database (1999–2011). *Investig. Ophthalmol. Vis. Sci.* **56**, 4719–4724 (2015).
10. Alexandrov, L. B. et al. Signatures of mutational processes in human cancer. *Nature* **500**, 415–21 (2013).

Reviewers' Comments:

Reviewer #1:

Remarks to the Author:

The authors have addressed my points. This is an exciting manuscript on a very important clinical unmet need in the melanoma field.

Reviewer #2:

Remarks to the Author:

Thank you for the clarification.

Reviewer #3:

Remarks to the Author:

I think the authors have replied satisfactory to the reviewer comments